# Microbiome characterization by high-throughput transfer RNA sequencing and modification analysis

Michael H. Schwartz[1,2], Haipeng Wang [1,3,4], Jessica N. Pan[1], Wesley C. Clark[1], Steven Cui[5],
Matthew J. Eckwahl[1], David W. Pan[1], Marc Parisien [1], Sarah M. Owens[5,6], Brian L. Cheng[2], Kristina Martinez[5],
Jinbo Xu[4], Eugene B. Chang[5], Tao Pan[1,2] & A. Murat Eren [2,5,7]

Advances in high-throughput sequencing have facilitated remarkable insights into the diversity and functioning of naturally occurring microbes; however, current sequencing strategies are insufficient to reveal physiological states of microbial communities associated with protein translation dynamics. Transfer RNAs (tRNAs) are core components of protein synthesis machinery, present in all living cells, and are phylogenetically tractable, which make them ideal targets to gain physiological insights into environmental microbes. Here we report a direct sequencing approach, tRNA-seq, and a software suite, tRNA-seq-tools, to recover sequences, abundance profiles, and post-transcriptional modifications of microbial tRNA transcripts. Our analysis of cecal samples using tRNA-seq distinguishes high-fat- and low-fat-fed mice in a comparable fashion to 16S ribosomal RNA gene amplicons, and reveals taxon- and diet-dependent variations in tRNA modifications. Our results provide taxon-specific in situ insights into the dynamics of tRNA gene expression and post-transcriptional modifications within complex environmental microbiomes.

[1] Department of Biochemistry and Molecular Biology, University of Chicago, Chicago, IL 60637, USA. [2] Committee on Microbiology, University of Chicago, Chicago, IL 60637, USA. [3] School of Computer Science and Technology, Shandong University of Technology, Zibo, Shandong, China. [4] Toyota Technological Institute at Chicago, Chicago, IL 60637, USA. [5] Department of Medicine, University of Chicago, Chicago, IL 60637, USA. [6] Argonne National Laboratory, Lemont, IL 60439, USA. [7] Marine Biological Laboratory, Woods Hole, MA 02543, USA. These authors contributed equally: Michael H. Schwartz, Haipeng Wang  Correspondence and requests for materials should be addressed to T.P. (email: taopan@uchicago.edu) or to A.M.E. (email: meren@uchicago.edu)

Microbes occupy almost all ecological niches on Earth[1], and the microbial communities associated with the human body play important roles in health and disease[2,3]. High-throughput sequencing of 16S ribosomal RNA (rRNA) gene amplicons and shotgun metagenomes enabled extensive culture-independent characterizations of microbial diversity and functional potential[4], however, these approaches are not suitable to gain insights into in situ physiology of microbial populations. An alternative way to study community structures of naturally occurring microbial populations, along with the added benefit of recovering information about thier physiological states, could rely on transfer RNA (tRNA) molecules.

A bacterial genome has 30–50 different tRNA gene sequences, and a single cell contains up to 100,000 tRNA transcripts[5]. Similar to the 16S rRNA gene, tRNA genes can be used as phylogenetic markers to resolve relationships between microbial taxa[6]. Yet, in contrast to the sequencing of 16S rRNA gene amplicons, which relies on PCR amplification from genomic DNA, the direct sequencing of tRNA transcripts (hereafter 'tRNA-seq') provides access to translational activities of the cell[7–11]. On average, a bacterial tRNA transcript contains 8 modifications, which play important roles in cellular physiology[12–14]. While modifications in the anticodon loop directly tune translational efficiency and fidelity, others tune tRNA stability and physical interactions with cellular proteins and the ribosome[15,16]. Systematical studies of bacterial tRNA modifications have so far been limited to five species in culture, *Escherichia coli*, *Bacillus subtilis*, *Mycoplasma capricolum*, *Lactococcus lactis*, and *Streptomyces griseus*[12,17–19], and used approaches such as two-dimensional thin-layer chromatography (TLC) and liquid chromatography-mass spectrometry (LC-MS). Insights from these studies revealed that while some tRNA modifications are shared among multiple taxa, others may be more unique. Taxon-dependent modifications may have distinct roles within complex microbial communities, and the identification of post-transcriptional tRNA modifications de novo could offer insights into changing metabolic activities of microbes as a response to changing environmental conditions. However, the presence of these modifications as well as the rigid structure of the tRNA molecule interrupt the cDNA synthesis that is required for sequencing, which limits comprehensive investigations of tRNA modifications and their utility to study microbial physiology in complex communities.

We recently addressed the experimental bottlenecks that limit the high-throughput sequencing of tRNA transcripts by employing two demethylase enzymes to remove tRNA modifications, and a thermophilic reverse transcriptase (TGIRT) to obtain long reads (demethylase tRNA-seq or DM-tRNA-seq[8]). In DM-tRNA-seq, each sample is split in two and processed with and without demethylase treatment prior to cDNA synthesis. We found that the demethylase-treated samples matched the cellular tRNA abundance measured by hybridization in human tRNA studies[8], whereas the untreated samples were well suited for the analysis of tRNA modifications[20]. TGIRT can read through many Watson–Crick base methylations while leaving behind mutation signatures that allow for the identification of certain modifications at single base resolution and enable comparative analyses of modification levels[20]. We have validated that the mutation fractions for the well-studied N1-methyladenosine (m1A) modification determined by tRNA-seq correlated well with modification fractions inferred by a gel-based method[20].

Here, we apply DM-tRNA-seq to gut microbiome of mice fed high-fat (HF) and low-fat (LF) diets and introduce an open-source software suite, tRNA-seq-tools, to identify and characterize tRNA sequences in high-throughput sequencing data that emerge from DM-tRNA-seq experiments. Our workflow enabled us to infer microbial community structures and taxonomy in our samples using tRNA sequences, investigate the expression of tRNAs with different anticodons across different environmental conditions, and recover tRNA modification levels across different taxa. Our results show class-dependent modification patterns, and diet-dependent changes in modification levels that can be associated with diet-dependent proteome in the microbiome. These findings demonstrate the versatility of tRNA-seq to investigate tRNA expression and modifications within microbiomes.

## Results

**tRNA-seq reveals tRNA modifications in bacterial cultures.** To determine the feasibility of employing tRNA-seq for modification studies in simple model systems, we performed DM-tRNA-seq on bacterial cultures from four species: *Escherichia coli*, *Bacillus subtilis*, *Staphylococcus aureus*, and *Barnesiella viscericola*. Bacterial tRNAs share some modifications with eukaryotic tRNAs, but also possess unique modifications of their own[21]. Since almost all *E. coli* tRNAs and the majority of *B. subtilis* tRNAs have been mapped for modifications by 2D-TLC and LC/MS, these two cultures allowed us to benchmark tRNA-seq results against the known modifications. In contrast, *S. aureus* and *B. viscericola* were suitable to identify new modifications using tRNA-seq.

We focused our analysis on Watson–Crick face base modifications for which DM-tRNA-seq was particularly well suited[20]. We first determined the mutation and the stop fraction at each nucleotide position within tRNA sequences[20]. Here, we calculated the mutation fraction by only including sequencing reads that pass the modification site while excluding shorter reads that stopped before the modification site. Figure 1 and Supplementary Figure 1 display representative mutation and stop fractions for several tRNAs with modification sites annotated using standard tRNA nomenclature[22]. *E. coli* tRNAPro(CGG) has N1-methylguanosine (m1G)37 and 4-thiouridine (s4U8), but only m1G can be removed by the demethylase treatment. As expected, both mutations and stops showed a reduction to the background levels at the m1G37 position in the demethylase-treated sample (Fig. 1a, Supplementary Fig. 1a). We identified a second mutation peak at position U8 that was more than 10-folds higher compared to the background level. This peak did not change in the demethylase-treated sample, consistent with the known s4U modification. *E. coli* tRNAPhe(GAA) has 3-(3-amino-3-carboxypropyl)uridine (acp3U)47, 2-methylthio-N6-isopentenyladenosine (ms2i6A)37, and s4U8, all were detected by mutation, and none responded to demethylase treatment (Fig. 1b, Supplementary Fig. 1b). Ms2i6A37 showed ~99% stops, and ~45% of the remaining 1% read-through were mutations. *B. subtilis* has m1A22 modifications, which are absent in *E. coli*. This methylation can be almost fully removed by the demethylases as reflected by the reduction of mutation fraction upon demethylase treatment for *B. subtilis* tRNASer(UGA) and tRNAGlu(UUC) (Fig. 1c, d). For tRNASer(UGA), the s4U8 was readily identified through its mutations (Fig. 1c), whereas the N6-isopentenyladenosine (i6A37) modification only showed stops (Supplementary Fig. 1c). Modifications in tRNAGlu(UUC) have not been previously mapped; our result is consistent with the presence of m1A22 in this tRNA (Fig. 1d, Supplementary Fig. 1d). No tRNA modifications have been mapped in *S. aureus* and *B. viscericola*. Our results are consistent with the presence of demethylase-sensitive m1G37 and m1A22 and demethylase-insensitive s4U8 in *S. aureus* tRNALeu(UAG) (Fig. 1e, Supplementary Fig. 1e), and m1A22 and s4U8 in tRNASer(GCU) (Fig. 1f, Supplementary Fig. 1f). Finally, we identified mutation and stop signatures that are consistent with the expected m1G37 and

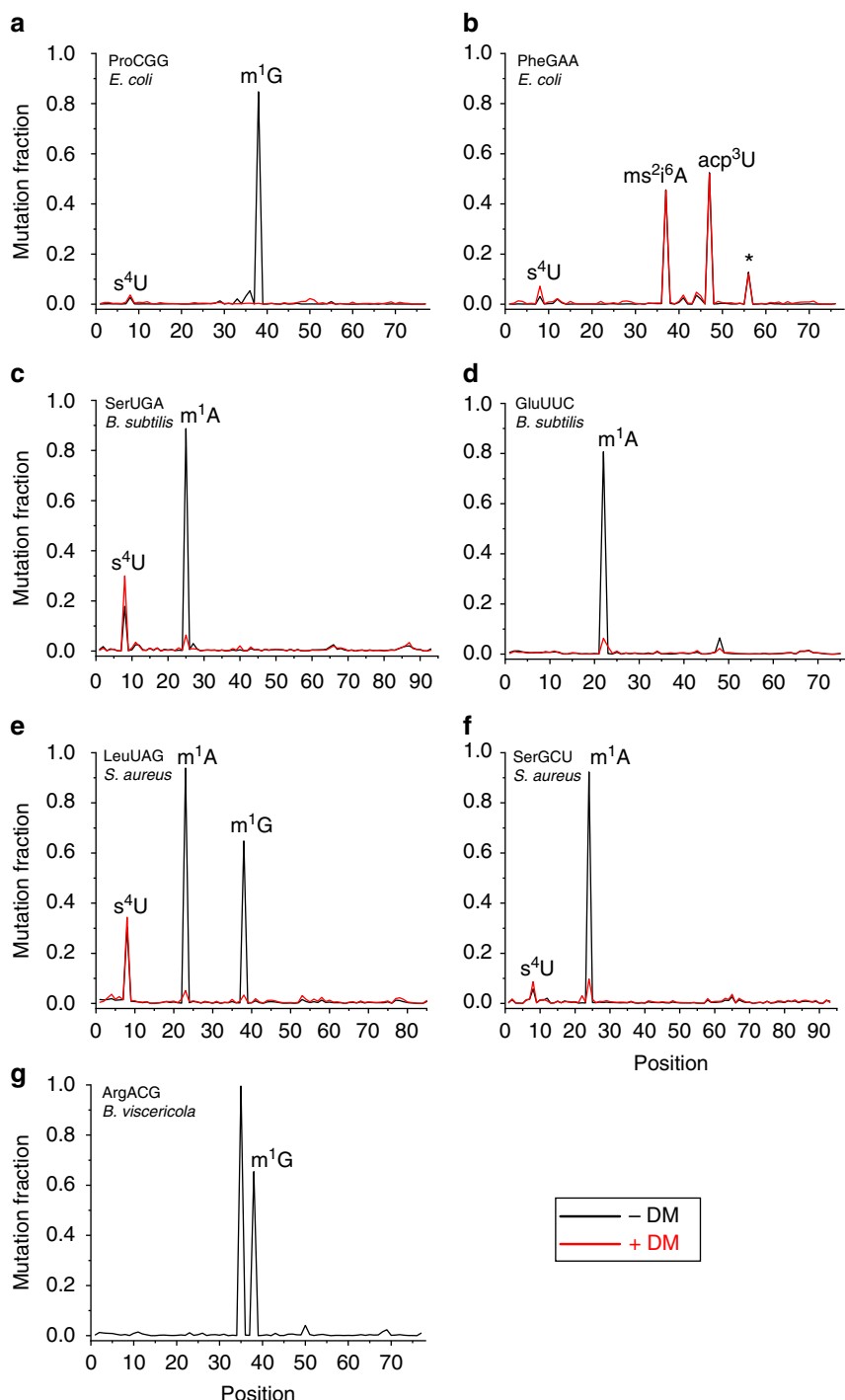

**Fig. 1** tRNA modifications of bacterial cultures. Red and black lines show mutation fractions in representative tRNA sequences with (+DM) and without (−DM) demethylase treatment, respectively. **a** *E. coli* tRNA$^{Pro}$(CGG) shows m$^1$G37 and s$^4$U8. **b** *E. coli* tRNA$^{Phe}$(GAA) shows acp$^3$U47, ms$^2$i$^6$A37 and s$^4$U8. The additional peak denoted by asterisk (*) may represent an unknown modification. **c** *B. subtilis* tRNA$^{Ser}$(UGA) shows m$^1$A22, and s$^4$U8. **d** *B. subtilis* tRNA$^{Glu}$(UUC) shows m$^1$A22. **e** *S. aureus* tRNA$^{Leu}$(UAG) shows m$^1$G37, m$^1$A22 and s$^4$U8. **f** *S. aureus* tRNA$^{Ser}$(GCU) shows m$^1$A22 and s$^4$U8. **g** *B. viscericola* tRNA$^{Arg}$(ACG) shows m$^1$G37 and I34

inosine (I)34 in *B. viscericola* tRNA$^{Arg}$(ICG) (Fig. 1g, Supplementary Fig. 1g). Using mutation and/or stop signatures, our results are consistent with all known Watson–Crick face base modifications in these four bacterial species (Supplementary Fig. 2, Supplementary Table 1).

We examined the complete mutation profiles that correspond to m$^1$A, m$^1$G, and s$^4$U for tRNAs in the four bacterial cultures (Fig. 2, Supplementary Fig. 3). We only found mutation signatures that are consistent with m$^1$A at position 22 located at the junction of the D stem and D loop in *B. subtilis* and *S. aureus*, but not in *E. coli* and *B. viscericola* (Fig. 2a). We found mutation signatures that are consistent with 13 and 12 tRNAs with m$^1$A22 modification in *B. subtilis* and *S. aureus*, respectively; only four have been previously mapped in *B. subtilis* by 2D-TLC[23,24]. Among all tRNAs with A22 in these two bacteria, all with G/A13–A22 pairs were modified, whereas all with U13–A22

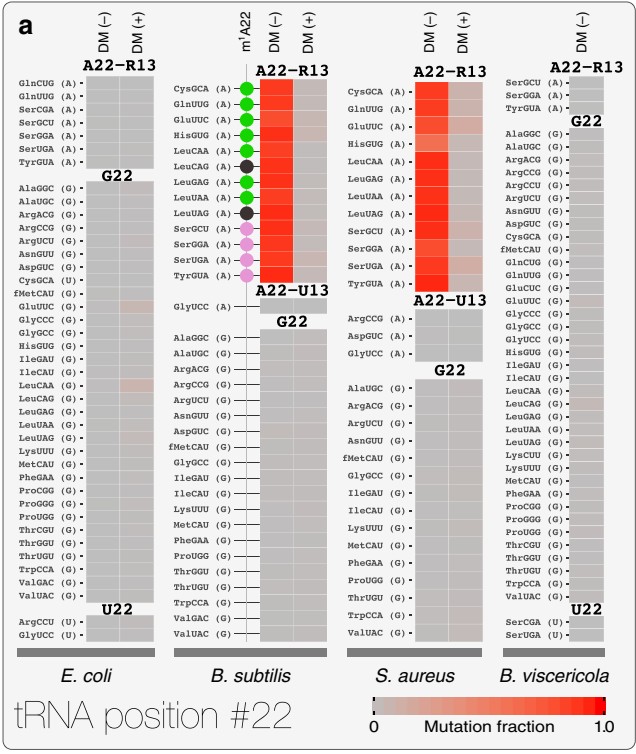
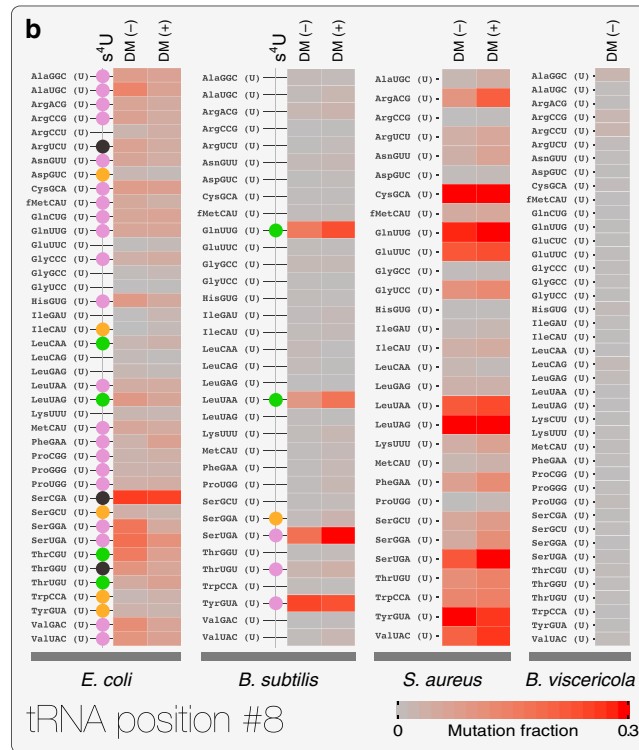

**Fig. 2** Mutation fractions of two tRNA sites. Heatmaps of mutation fractions for positions 22 and 8 (using standard, canonical tRNA nomenclature) are shown. tRNAs with different anticodons are grouped by their sequences at the respective position of modification (in parenthesis) and in alphabetical orders. Only *E. coli* and *B. subtilis* tRNA modifications have been mapped previously by 2D-TLC and LC/MS, but the mapping was not done for every tRNA species[12]. Every *E. coli* and *B. subtilis* tRNA species with mutation fraction at10-times above background is marked with a circle on the right with the following designations: Purples correspond to those known to be present and also identified by sequencing here; blacks correspond to those supposed to be absent but identified by sequencing; greens correspond to those not mapped previously but identified by sequencing; oranges correspond to those known to be present but were not found by sequencing. **a** $m^1A22$; R corresponds to A or G. **b** $s^4U8$

were not modified, consistent with this modification requiring a non-Watson–Crick base paired A22. $M^1A22$ mutation fractions ranged from 0.81–0.94 in *B. subtilis* and 0.61–0.94 in *S. aureus*, suggesting that most tRNAs are modified at high fractions at A22 under our culture conditions. $M^1G$ is located at position 37 immediately 3′ to the anticodon nucleotides in bacteria. In all four bacteria, we found that mutation fractions ranged from 0.33–0.90 when the tRNA has G37 in its sequence (Supplementary Fig. 3). Unlike $m^1A22$, $m^1G37$ was accompanied by highly variable stop fractions in a tRNA-dependent manner (ref. [20], Supplementary Fig. 1). Since $m^1G37$ prevents translational frameshifting[25,26] and is thought to be present at high fractions in tRNA, mutation alone may not correlate well with modification fractions for $m^1G37$.

We identified mutation signatures consistent with many $s^4U8$ modifications in three of the four bacteria (Fig. 2b). We found 28 sites in the *E. coli* tRNA with mutation fractions of 0.03–0.26 (0.03 corresponded to >10× above background), 20 of which overlapped with the known $s^4U8$ modifications by 2D-TLC and LC/MS. The other 8 sites could either be present only under our culture conditions or were missed in the previous mapping studies. We found 5 sites in *B. subtilis* tRNAs with mutation fractions of 0.03–0.25, two of which overlapped with the known $s^4U8$ modifications. We found 21 sites with mutation fractions of 0.03–0.39 in *S. aureus* tRNAs, indicating that *S. aureus* tRNAs can be highly modified with $s^4U8$. *B. viscericola* showed no mutation significantly above background for any tRNA, suggesting that $s^4U8$ is absent in this bacterium. Overall, our results suggest that bacterial tRNA modifications can be highly variable

at the transcriptome level among tRNA species even under mid-log growth conditions. These findings all together demonstrate the utility of tRNA-seq to study modifications in bacterial tRNA.

**Community and anticodon profiles in gut microbiomes.** Next, we evaluated the utility of tRNA-seq to infer tRNA abundance and modification profiles in complex gut microbial ecosystems using mice that were fed either a HF or LF diet. Since HF and LF diet affect microbial community structures in the gut[27], this model provided an opportunity to also compare community structures inferred by tRNA-seq to those that were identified through 16S rRNA gene amplicons. We collected our samples from the mouse cecum, and processed them both for 16S rRNA gene analysis for taxonomic comparisons, and tRNA-seq with and without demethylase treatment. We developed an open-source software, tRNA-seq-tools (https://github.com/merenlab/tRNA-seq-tools) to de novo identify tRNA sequences with conserved sequence motifs and secondary structures from the raw sequencing data (Fig. 3a). We assigned taxonomy to resulting tRNA sequences, and we used only those sequences that were assigned to a unique anticodon for downstream analyses.

DM-tRNA-seq generated all tRNA reads starting from the conserved 3′-CCA[8]. We designed tRNA-seq-tools to use the canonical tRNA sequence motifs (Supplementary Fig. 4) that start with the 7 nucleotides 3′ acceptor stem and 17 nucleotides TΨC stem-loop, a variable region of 4–5 for type I and 13–22 nucleotides for type II tRNAs (tRNA^Ser/Leu/Tyr for bacteria), and

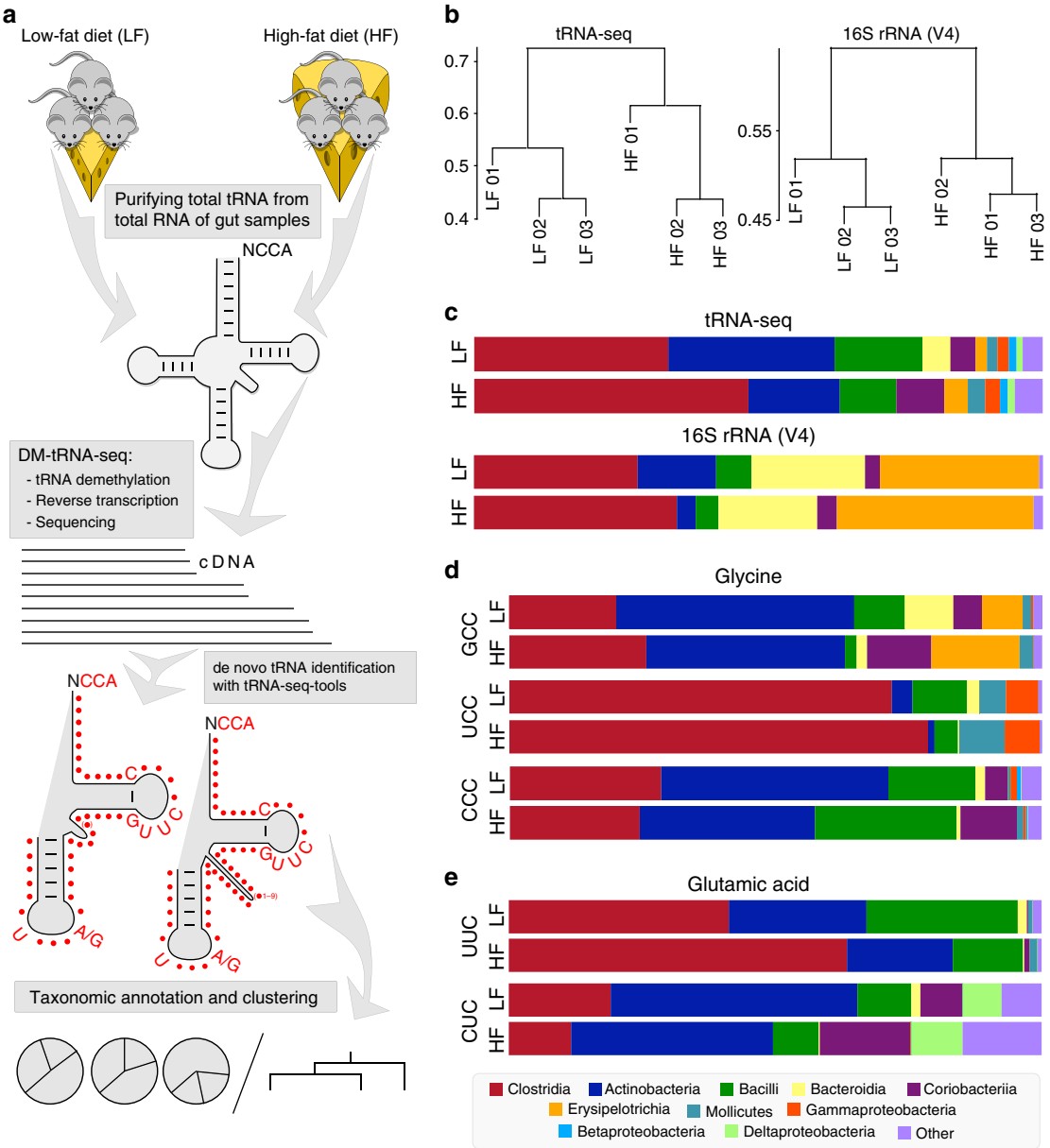

**Fig. 3** Microbiome tRNA-seq workflow and taxonomy analysis. **a** Workflow of tRNA sequencing of gut microbiome samples fed with a high-fat (HF) or low-fat (LF) diet and de novo tRNA assignment. Conserved tRNA residues that were searched for in this work are shown in red. **b** Dendrograms compare relationships between HF and LF samples that were inferred based on community profiles of tRNA transcripts, or 16S rRNA gene amplicons. **c** Class-level taxonomy for averaged HF and LF samples based on tRNA-seq (top) and 16S rRNA gene amplicons (bottom). All bacterial classes at >1% level are shown in distinct colors, all other bacterial classes are grouped together and shown in purple. **d** tRNA$^{Gly}$ taxonomy for anticodons GCC, UCC, and CCC. **e** tRNA$^{Glu}$ taxonomy for anticodons UUC and CUC. Among the other category for GCC/UCC/CCC and UUC, no class has an abundance of ≥1%; for CUC, other classes with an abundance of ≥1% include Alphaproteobacteria, Gemmatimonadetes, and Ignavibacteria. tRNAs decoding these two amino acids are the most abundant in our tRNA-seq results

finally a 17 nucleotides anticodon stem-loop. We included the conserved, canonical tRNA sequence of GTTC/C in the TΨC stem-loop and T 5′ and A/G 3′ to the anticodon nucleotides. We further restricted our assignment to anticodon stems containing only Watson–Crick or G-U wobble base pairs, but allowed for one mismatch among all conserved nucleotide sequences. The minimal length of our assigned tRNA reads corresponded to the 5′ end of the anticodon stem to the 3′-CCA which was 48–49 for type I and 57–66 nucleotides for type II tRNAs.

Our analysis on average resulted in ~2.2 million tRNA sequences per demethylase-treated sample containing unambiguous anticodon assignments (Supplementary Table 2). There

were on average ~3.8 million other reads that were either too short to be assigned to a unique tRNA anticodon or did not unambiguously fit to the canonical tRNA signatures. Short sequences are often the result of persistent modifications demethylases fail to remove, or other tRNA structural features that interrupt the cDNA synthesis. An example of the former in our data was the ms$^2$i$^6$A37 modification in *E. coli* tRNA$^{Phe}$ which stopped the RT at ~99% (Supplementary Fig. 1b). Therefore, most tRNA$^{Phe}$ reads in *E. coli* did not reach the anticodon and would not have been taken into consideration here. For the tRNA sequences with anticodon assignments, grouping the biological replicates by individual anticodons showed good compatibility

among the HF or LF groups (Supplementary Fig. 5). Our tRNA extraction method from the mouse cecum resulted in less than 1% contamination with mouse tRNA sequences, which was consistent with our microarray studies[28]. We randomly selected one type I and one type II tRNA and aligned matching quality-filtered short reads to visualize the coverage and homogeneity of individual nucleotide positions (Supplementary Fig. 6). Consistent with our assignment rules, all reads were beyond the 5' stem of the anticodon stem-loop with gradual drop-offs toward the 5' end of tRNA, which we had also observed with human tRNA reads[8].

We performed de novo analysis of tRNA sequences and 16S rRNA gene amplicons using Minimum Entropy Decomposition[29] to infer community structures, which revealed nearly identical relationships between samples (Fig. 3b). We also compared the average taxonomic distribution of HF/LF samples from tRNA-seq and 16S rRNA gene amplicons (Fig. 3c). While we used the SILVA database[30] with >350,000 entries to assign taxonomy for 16S rRNA gene amplicons, we used 4235 gold-standard bacterial genomes obtained from the Ensembl database[31] to assign taxonomy for tRNA sequences (Supplementary Table 3). For the six most abundant bacterial classes, the full tRNA taxonomy qualitatively matched the 16S rRNA gene-based taxonomy (Fig. 3c), but proportions of taxa differed between the two approaches. For instance, tRNA-seq showed higher fractions of Clostridia and Actinobacteria, and lower fractions of Bacteriodia and Erysipelotrichia classes. Multiple factors could result in differences between abundance estimates of the two approaches, one among them being the utilization of RNA transcripts in the tRNA-seq workflow in contrast to PCR amplification of genomic DNA in the 16S rRNA gene-based workflow (see Discussion).

We also analyzed the taxonomic make up of tRNA sequences at the anticodon-level (Fig. 3d, e, Supplementary Table 4). The two most abundant tRNA sequences in our dataset decoded amino acids of glycine and glutamic acid. Previous analyses based on available genomes show that for tRNA$^{Gly}$ (Fig. 3d), UCC is present in only all, GCC in most, and CCC in only a small number of bacterial taxa[32]. The taxonomy of tRNA$^{Gly}$(GCC) was similar to the taxonomic make up of all tRNA sequences. The tRNA$^{Gly}$(UCC) taxonomy has a larger representation of Clostridia, and the tRNA$^{Gly}$(CCC) taxonomy diverged from the all-tRNA taxonomic profiles such as it showed more Bacilli in the HF sample. Similarly, for tRNA$^{Glu}$ (Fig. 3e), UUC is present in all, while CUC is found only in some bacterial genomes. Our results showed that the tRNA$^{Glu}$(UUC) taxonomy was more similar to the overall taxonomy of all tRNA sequences, whereas the tRNA$^{Glu}$(CUC) taxonomy had a different profile with a larger representation of Actinobacteria in the LF sample.

To gain further insights into tRNA anticodon-based taxonomy, we performed additional analyses of the anticodons CUC and UUC of sequences that encode for glutamic acid (Supplementary Fig. 7). tRNA$^{Glu}$(UUC) can read both GAA and GAG codons and is essential in all cells, whereas tRNA$^{Glu}$(CUC) can only read GAG codon and is optional for life. We found that the taxonomic proportion of tRNA$^{Glu}$(UUC) varies widely among the six major bacterial classes compared to the taxonomic profiles based on all tRNA sequences as well as 16S rRNA gene amplicons (Supplementary Fig. 7a). Zooming into the class Bacilli further, we found that 75% of all tRNA$^{Glu}$(CUC) genes (78/104) are concentrated in the Lactobacillaceae family (Supplementary Fig. 7b), suggesting that this family can be uniquely assessed using tRNA$^{Glu}$(CUC) reads. Lactobacillaceae could be resolved both through tRNA sequences and 16S rRNA gene amplicons which showed similar abundance patterns between HF and LF microbiome samples (Supplementary Fig. 7c).

Focusing further on tRNA$^{Glu}$(CUC) and tRNA$^{Glu}$(UUC) reads from the family of Lactobacillaceae, we found that the ratio of tRNA$^{Glu}$(CUC) to tRNA$^{Glu}$(UUC) was higher in the LF samples than the HF samples (Supplementary Fig. 7d). This could be due to either higher tRNA$^{Glu}$(CUC) expression or a higher proportion of Lactobacillaceae organisms that contain tRNA$^{Glu}$(CUC) in the LF samples. In the case of the former, the increased tRNA$^{Glu}$(CUC) expression could potentially add an additional capacity for the Lactobacillaceae in LF samples to specifically decode the GAG codon in translation. Finally, we found that all tRNA$^{Glu}$(CUC) reads in our samples are from the Lactobacillus genus in the Lactobacillaceae family. This genus is represented by 31 genomes in our tRNA database; 18 different tRNA$^{Glu}$(CUC) gene sequences are present among the 31 genomes, which allowed us to investigate the species-level distribution of reads that resolved to tRNA$^{Glu}$(CUC) (Supplementary Fig. 7e). Overall, these results indicate that anticodon-level taxonomic analyses of tRNA transcripts can provide additional insights into physiological differences between individual branches of bacteria.

**Diet dependent differences in tRNA modification levels.** A unique feature of tRNA-seq is the capability to assess in situ tRNA modifications. Compared to bacterial cultures, the analysis of modifications in tRNA transcripts in complex microbial communities presents additional challenges. For instance, even though both mutations and stops are useful for tRNA modification studies in pure cultures, in microbiome studies we can only rely on mutation information since stops yield short reads that often do not assign to specific tRNA seed sequences unambiguously. Mutations in the sequencing data can be identified through the alignment of individual reads to reference seed sequences. For bacterial cultures, the seed sequences are simply the annotated tRNA genes. In contrast, complex environments may require de novo-identified tRNA sequences to serve as seed sequences for modification analyses due to the lack of comprehensive reference genomes (Fig. 4a, Supplementary Fig. 8).

To study modification fractions we selected our seed sequences from demethylase-treated samples. Our clustering-based approach (see Materials and Methods) resulted in an average of ~10,700 seed sequences for each sample, which was approximately proportional to the total number of tRNA reads per sample (Supplementary Table 2).

We ran a multiple sequence alignment for the same anticodon while allowing flexible lengths in the variable loop and the α and β regions of the D loop to visualize our results. Figure 4 shows our workflow to study modification fractions, and the alignment results for the tRNA$^{Ser}$ transcripts in a HF sample. We found three major peaks for tRNA$^{Ser}$(GCU) (Fig. 4b) without demethylase treatment. Peaks 2 and 3 disappeared upon demethylase treatment, therefore assigning them to m$^1$A. Peak 1 remained upon demethylase treatment, thereby assigning it to s$^4$U. We found the same 3 peaks for the other three tRNA$^{Ser}$s with the same responses to demethylase treatment (Fig. 4c–e). We found a demethylase-removable fourth peak for tRNA$^{Ser}$(CGA) (Fig. 4e) which may represent m$^1$A in tRNA species with shorter variable arms.

We then analyzed bacterial taxon-dependent modifications. Examples in Fig. 5a were tRNA$^{Ser}$(UGA) in the same microbiome sample. For this tRNA from Lactobacillus, class Bacilli, we readily identified mutation signatures consistent with m$^1$A22 and s$^4$U8 (Fig. 5a, Supplementary Fig. 9a), both were also known in tRNA$^{Ser}$(UGA) of B. subtilis, class Bacilli. For Bifidobacterium, class Actinobacteria, mutation signature consistent with an m$^1$A in the T loop (m$^1$A59 in standard tRNA nomenclature) can be readily identified, but this tRNA did not have s$^4$U8 (Fig. 5a,

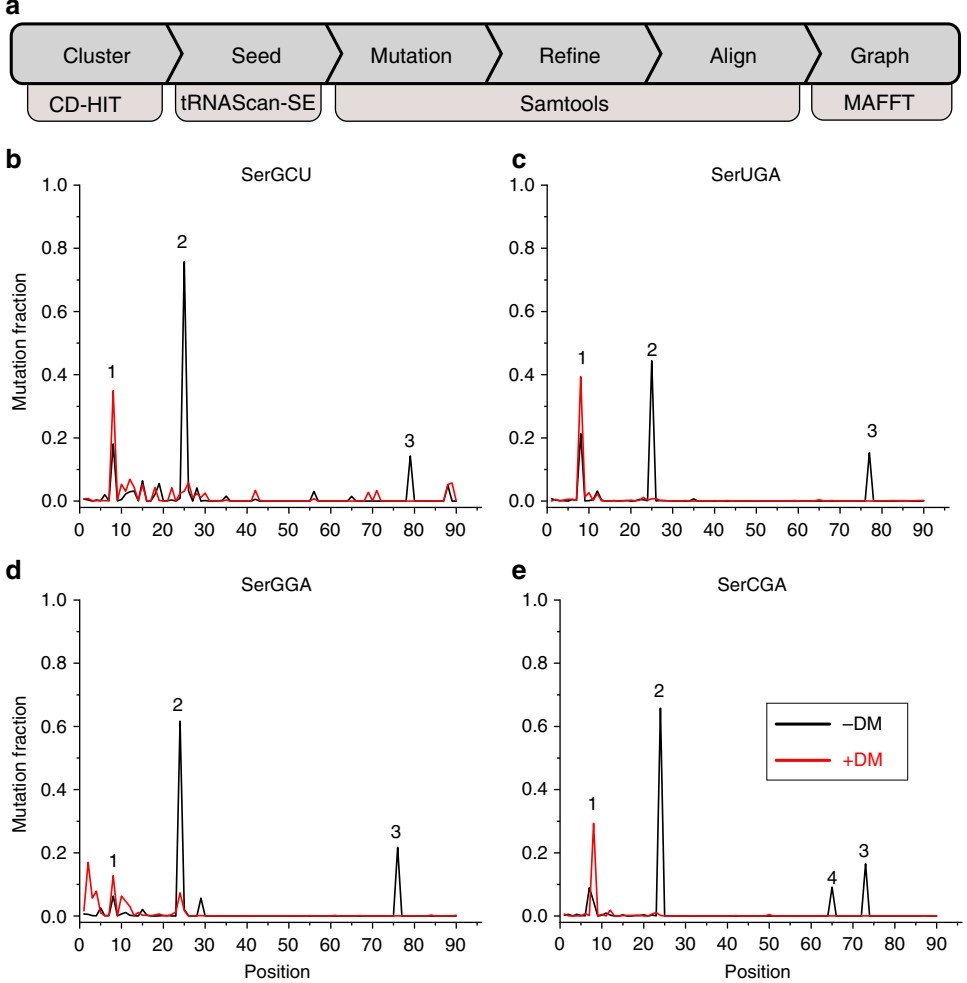

**Fig. 4** Microbiome tRNA modification analysis. **a** Workflow for modification assignment using mutation signatures. **b**–**e** Representative positional plots showing m$^1$A and s$^4$U modifications for transcripts of tRNA$^{Ser}$(GCU) (**b**), tRNA$^{Ser}$(UGA) (**c**), tRNA$^{Ser}$(GGA) (**d**), and tRNA$^{Ser}$(CGA) (**e**), HF-fed mouse sample. The peak numbers correspond to those described in the text with peak 1 called for s$^4$U8 and peak 2 for m$^1$A22. Peak 3 is located around nucleotide 73–79 in the type II tRNA$^{Ser}$s, but is m$^1$A59 in the standard tRNA nomenclature. Red and black lines show mutation fractions in tRNA$^{Ser}$ sequences with (+DM) and without (−DM) demethylase treatment, respectively

Supplementary Fig. 9b). M$^1$A59 modification has also been found in *Streptomyces griseus*[12] in class Actinobacteria, although *S. griseus* is in the order Actinomycetales, whereas *Bifidobacterium* belongs to the order Bifidobacteriales. We also identified the more common m$^1$A58 modifications, e.g. in tRNA$^{Phe}$ from *Bifidobacterium* (Supplementary Fig. 9c).

We identified a class-dependent modification pattern at three positions (Fig. 5b). M$^1$A22 was predominantly present in the closely related Clostridia and Bacilli classes. M$^1$A58 and m$^1$A59 were predominantly present in Actinobacteria. S$^4$U8 is widely spread in most of the bacterial classes, but absent in Bacteriodia and Deltaproteobacteria. The absence of m$^1$A and s$^4$U in Bacteriodia is unlikely due to the lack of sequencing depth as this result is consistent with our bacterial culture study that did not find s$^4$U8 for *B. viscericola* from class Bacteriodia (Fig. 2b). We superimposed the three m$^1$A modifications onto the three-dimensional tRNA structure (Fig. 5c). M$^1$A22, m$^1$A58, and m$^1$A59 all introduce a positive charge in the tRNA, and are all located in the elbow region of the tRNA structure, suggesting that they may serve a common function.

To investigate whether modification fractions changed between HF and LF group, we analyzed the mutation fractions of m$^1$A and s$^4$U sites in all tRNAs (Fig. 6). The mutation fraction is not fully quantitative for absolute modification fraction due to tRNA sequence context-dependent differences. However, comparative analyses of modification levels at each site between two samples can be interpreted with high confidence in regard to relative changes in modification fractions, as each site has the same sequence context. For Lachnospiraceae, m$^1$A22 was present in eight tRNA anticodon groups, but in seven of the eight, the median level was higher in the HF than the LF samples (Fig. 6a). For Bifidobacteriaceae, m$^1$A58 and m$^1$A59 were present in 12 tRNA anticodon groups; in all 12 groups, the median level was higher in the HF than the LF samples (Fig. 6b). The m$^1$A level change under dietary conditions of the mouse gut microbiome may be related to translation activity. In a human cell study, m$^1$A modified tRNA had an increased affinity to the translation elongation factor and increased the expression of a reporter gene[33]. On the other hand, m$^1$A-hypomodified tRNAs may be better tuned for interaction with other proteins in cells to perform extra-translational functions[34].

For Lachnospiraceae, s$^4$U8 was present in 24 tRNA anticodon groups; no clear preference for HF versus LF samples could be discerned (Fig. 6c), indicating that the s$^4$U levels did not respond to the dietary conditions of the mouse gut microbiome. S$^4$U8 is known to serve as a sensor to elicit cellular responses to UV;[35] gut

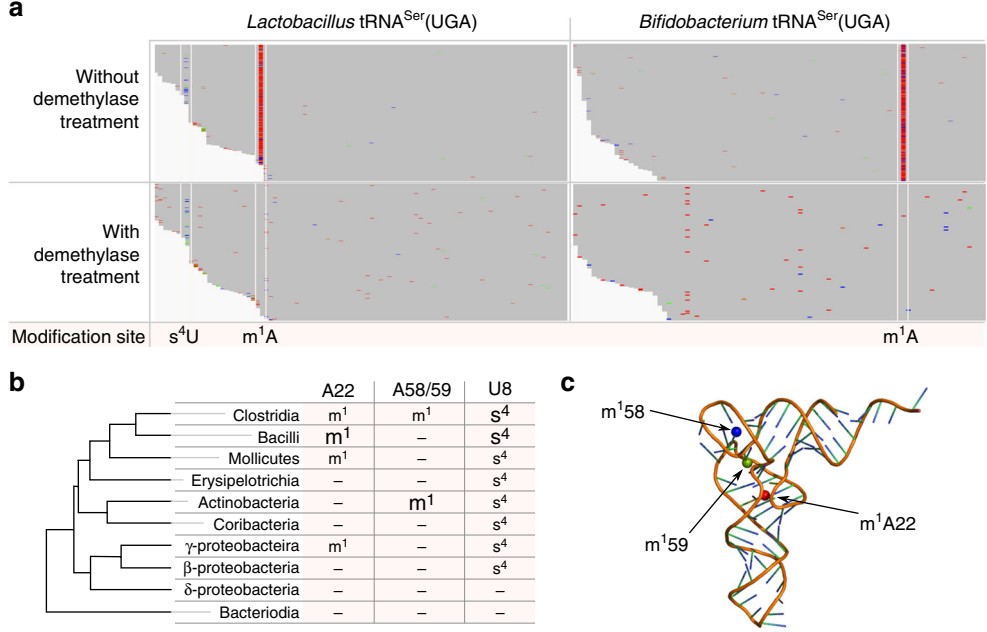

**Fig. 5** Taxonomic differences of modification sites. **a** Examples of aligning tRNA sequencing reads to two seed sequences of tRNA$^{Ser}$(UGA) from *Lactobacillus*, class Bacilli, and *Bifidobacterium*, class Actinobacteria without and with demethylase treatment. Modification sites identified (s$^4$U and m$^1$A) are highlighted between the white lines. **b** m$^1$A22, m$^1$A58/59, and s$^4$U8 identified in the abundant bacterial classes from Fig. 3c in the context of their phylogenetic relationship. Large fonts indicate bacterial classes in which the majority of the modifications are found (m$^1$A22 in Clostridia and Bacilli, m$^1$A58/59 in Actinobacteria, and s$^4$U8 in Clostridia and Bacilli). **c** Proximal location of the m$^1$A22 (red), m$^1$58 (blue), and m$^1$59 (green) modifications in a tRNA three-dimensional structure

microbiomes are not exposed to UV light and therefore may not adjust its s$^4$U8 levels to dietary differences.

**tRNA modification fractions and protein expression levels.** While differences across dietary conditions suggest tRNA modification fractions could reflect environmental responses, these results by themselves do not communicate whether modification fractions are associated with protein synthesis. To investigate associations between the m$^1$A modification fractions observed in our study and protein synthesis, we used the gut metaproteomics data generated by Figeys, Stintzi, Mack, and co-workers[36] also from mice fed with HF and LF diet. In their study, the authors describe a total of 849 proteins with significantly different expression levels between the HF and LF samples collected over a course of 43 days. The day 29 and 43 experimental conditions most closely resemble the experimental condition of our tRNA-seq experiment. The metaproteomics data from both day 29 and 43 showed a marked difference between HF and LF conditions (e.g., Figure 7a). Our taxonomic analysis assigned 77% (655/849) of these proteins to the class Clostridia (Fig. 7b), and we focused our subsequent analysis specifically on this group (Fig. 7c). To examine a possible relationship between these differentially expressed proteins and tRNA m$^1$A modifications, we compared the amino acid and codon compositions of these proteins that are highly expressed in HF samples (log >1) and depleted in HF samples (log < −1). In Clostridia, m$^1$A modification is present in tRNAs that decode amino acids Cys, Glu, Gln, and Ser (Fig. 6a). We combined amino acid residues or codons based on protein expression levels in HF over LF, and subtracted the compositions of HF over-expressed proteins minus the HF under-expressed proteins (Fig. 7d, e). In both cases, Cys and Glu, but not Gln and Ser are over-represented among the highly over-expressed proteins, although this over-representation was not unique to Cys and Glu.

Since ribosome decoding includes an adjacent pair of mRNA codons in the A and P site, we also combined amino acid and codon pairs in a similar fashion (Fig. 7f, g, Supplementary Fig. 10). Excitingly, the top five most over-represented amino acid pairs in both day 29 and 43 HF over-expressed proteins relative to the HF depleted proteins were E-E (Glu-Glu), K-E (Lys-Glu), L-E (Leu-Glu), E-K (Glu-Lys), and R-E (Arg-Glu). For codon pairs, the top five most over-represented were GAG-GAA (Glu-Glu), AAG-GAA (Lys-Glu), GCG-GAG (Ala-Glu), CTG-GAA (Leu-Glu), and CCG-GAG (Arg-Glu) which closely matched the over-represented amino acid pairs. Decoding the top five over-represented amino acid and codon pairs all involves the m$^1$A-containing tRNA$^{Glu}$ (anticodon CUC, UUC) in HF samples (Fig. 6a). These proteomic results are therefore consistent with our hypothesis that tRNAs with higher m$^1$A levels enhance the decoding of their respective amino acid/codon pairs. Overall, this analysis indicates that tRNA m$^1$A modification level differences are consistent with m$^1$A modified tRNAs facilitating increased expression of microbiome proteins with specific amino acid and codon contexts.

## Discussion

Here we described an approach to perform and characterize the high-throughput sequencing of tRNA transcripts that enables insights into the translational states of naturally occurring microbial communities. Although we previously have demonstrated the efficacy of tRNA-seq for studying human cells, the application of this strategy to study environmental microbes comes with unique experimental and computational challenges that have not been addressed previously.

Our study establishes a computational workflow for the recovery of tRNA sequences and the characterization of their abundance and modification patterns in environmental microbiomes. However, there is substantial room for further

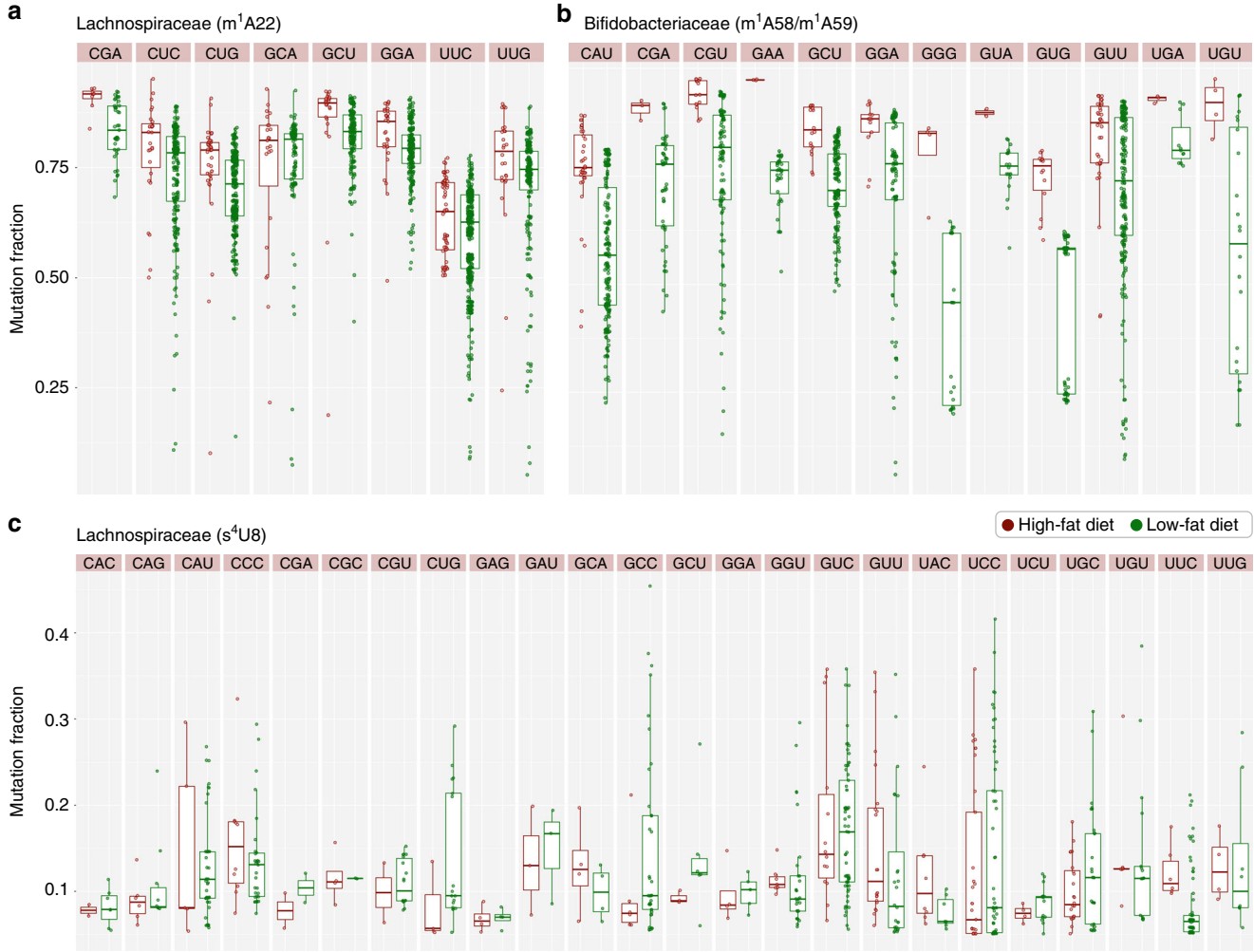

**Fig. 6** Comparisons of mutation fractions of HF versus LF samples. Bacterial families with the highest numbers of modifications at the respective nucleotides are shown. Each pair shows HF and LF samples with distinct anticodons marked on top. The amino acid whose codons are read by the corresponding tRNA with designated anticodon can be found in Supplementary Table 4. Box-and-whisker plots show median as a line, upper and lower quartiles in the box, and outliers outside of the line. **a** m$^1$A22 from Lachnospiraceae, class Clostridia. **b** m$^1$A58 and m$^1$A59 from Bifidobacteriaceae, class Actinobacteria. **c** s$^4$U8 from Lachnospiraceae, class Clostridia

improvements. For example, the tRNA gene database we used for taxonomic annotations contained only gold-standard isolate genomes of bacteria. tRNA gene annotation from the large number of metagenome-assembled and single-amplified genomes[37] could dramatically expand the database of reference tRNA sequences. This would be particularly important for applying tRNA-seq to ecological samples for which fewer isolate genomes are available. Our strategy to de novo identify tRNA sequences of identical origin to infer microbial community structures relied on an algorithm which has originally been developed for marker gene amplicon surveys[29,38]. However, unlike the sequencing products of most marker gene amplicon surveys, tRNA sequences contain more length variation. Hence, potential improvements of recovering abundance and distribution patterns through more specific algorithms warrant further study. Finally, strategies to select seed sequences for modification analyses directly influence the de novo identification of tRNA modification sites and mutation rates in environmental samples. While our current workflow only relied on seed sequences recovered from tRNA-seq data itself, this critical step could benefit from more comprehensive approaches that combine de novo identified sequences from tRNA-seq with the expanded reference database of tRNA gene sequences.

On the experimental side, bacterial tRNA modifications at the Watson–Crick face that cannot be removed by the demethylases can bias the quantitative analysis of tRNA-based taxonomy. This is particularly difficult for certain tRNA families with modifications 3' to the anticodon that stop the thermophilic reverse transcriptase we used here. For example, the ms$^2$i$^6$A37 modification seems to be present in tRNA$^{Phe}$ of three cultured bacteria we studied here. Because our current microbiome analysis assigns each tRNA to a specific anticodon, any read that stopped at this position would not have an anticodon assignment. Such modification-dependent RT stops can explain in part why on average only ~33% of tRNA reads were assigned to a unique anticodon in this work. One possible improvement is to chemically or enzymatically remove this modification from microbiome tRNA prior to cDNA synthesis. The relatively high biomass requirement, which is in the range of 0.1–10 μg total RNA, of the current experimental workflow poses another bottleneck. While this limitation is not necessarily a concern for gut microbiome studies, most applications of tRNA-seq to terrestrial and marine habitats with low microbial biomass will require improvements in molecular biology techniques.

Our analyses that used the same set of samples both with 16S rRNA gene amplicons and tRNA-seq revealed quantitative

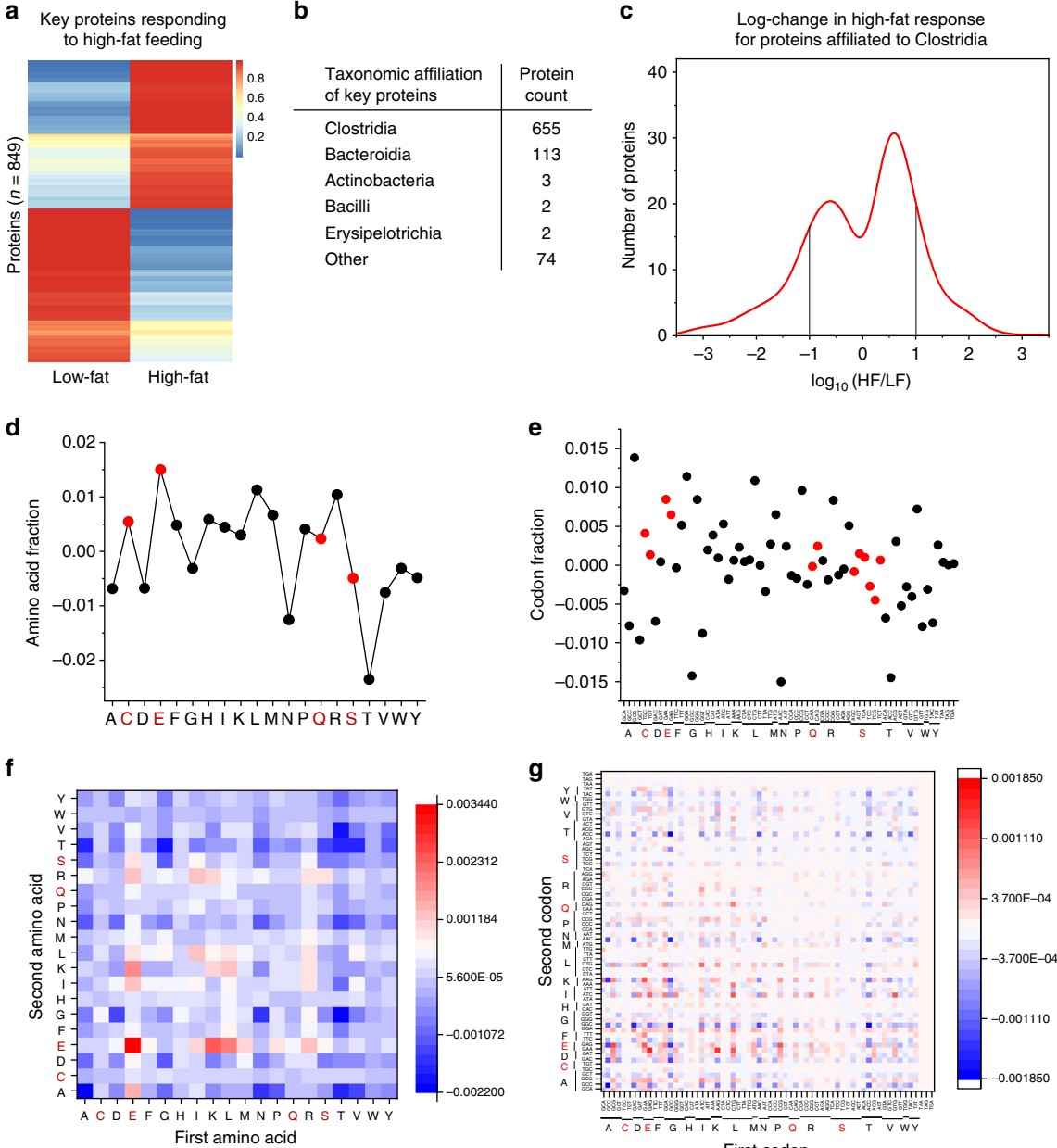

**Fig. 7** Analysis of differential protein expression and tRNA modification. Proteomics data from HF- and LF-fed mice were from reference[36]. **a** Average differential expression of 849 proteins between HF- and LF-fed mouse gut microbiome from day 43 mice that most mimics the experimental condition of our tRNA-seq experiment. **b** BLASTp protein sequence analysis shows that most of these proteins are from class Clostridia. **c** Quantitative difference between the clostridia proteins from day 43 mice. Lines show the boundaries of the proteins used for downstream analysis that are highly enriched (log >1, 88 proteins) or depleted (log< −1, 105 proteins) in HF over LF samples. The difference in amino acid (**d**) or codon content (**e**) determined by subtracting the compositions of HF over-expressed proteins minus the HF under-expressed proteins. The amino acids or codons for which their decoding tRNAs were found to contain m$^1$A modifications are in red: Cys, Glu, Gln, Ser. The difference in amino acid pair (**f**) and codon pair content (**g**) determined by subtracting the pair compositions of HF over-expressed proteins minus the HF under-expressed proteins. The first amino acid represents the N-terminal residue and the first codon represents the 5′ codon

differences in estimates of taxonomic composition, which may be due to several factors. For instance, the taxonomic analysis of tRNA sequences in our study relied on a database of ~4000 isolate genomes with biases towards certain bacterial groups. In contrast, the same analysis for 16S rRNA gene amplicons used a database of >350,000 rRNA gene sequences. In addition, RT-stopping modifications can also bias tRNA abundances in a taxa-dependent manner. It is also worth noting that in contrast to the tRNA-seq workflow, the 16S rRNA gene amplicons originate from the genomic DNA through the use of 'universal' PCR

primers, the efficiency of which may differ among microbial branches[39,40], and can amplify environmental DNA from inactive or dead cells[41]. To what extent these factors are responsible for the observed differences and how they impact our understanding of microbial ecosystems need further investigation.

Besides these challenges, tRNA-seq comes with several distinct strengths as it offers access to an overall physiological assessment of all cells in a complex environmental sample through an actively expressed gene product via high-throughput sequencing. As it does not require any prior knowledge, tRNA-seq enables

investigations of tRNA modifications associated with uncharacterized bacterial branches or tRNAs with non-canonical sequence and structural features. For instance, our application of tRNA-seq to mouse gut microbiome revealed previously unknown class-dependent tRNA modification patterns in bacteria. While $m^1A22$ and $s^4U8$ in Bacilli are the same as those in *B. subtilis*, here we were able to map the presence or absence of $m^1A22$, $m^1A58$, $m^1A59$, and $s^4U8$ in other bacterial classes. The class-dependent modification patterns at these locations suggest a non-identical use of tRNA modifications. Unexpectedly and excitingly, the $m^1A$ sites showed significant differences in their modification levels for the gut microbiome fed with different diets. Our analysis of amino acid and codon contents of the metaproteomes under the same dietary conditions lend support to our hypothesis that increased $m^1A$ modification levels enhance translation of specific proteins in the microbiome in an amino acid and codon context-dependent manner. These results suggest a potential use of modification fractions as molecular markers to investigate microbial responses to changing environments.

In summary, we described an approach that makes use of high-throughput sequencing of tRNA transcripts to gain insights into physiological states of naturally occurring microbial communities. tRNA-seq represents another direction in microbiome research as it relies on an actively expressed gene product that is core to protein synthesis machinery, universally abundant, and phylogenetically informative. It offers a complementary approach to the sequencing of 16S rRNA gene amplicons and genome-resolved metagenomics for broader characterization of environmental microbiomes.

## Methods

**Bacterial culture growth**. We grew *E. coli* BW25113, *B. subtilis* 168, and *S. aureus* RN4220 cultures at 37 °C in Erlenmeyer flasks with a 10:1 flask medium ratio at 225 rpm in a shaker. We grew *E. coli* and *B. subtilis* in LB Lennox and *S. aureus* in tryptic soy broth. We obtained *Barnesiella viscericola* C46 from the Deutsche Sammlung von Mikroorganismen und Zellkulturen (DSMZ). We grew *B. viscericola* in serum vials under strict anaerobic conditions at 37 °C in chopped meat medium[42]. We harvested all cultures at midlog by centrifugation.

**Animals**. All murine experimental procedures were approved by the University of Chicago Institutional Animal Care and Use Committee (IACUC). We followed all relevant ethical regulations. We bred C57BL/6 mice and maintained them under standard 12:12 h light/dark conditions at the University of Chicago. We fed age and litter-matched specific pathogen free (SPF) male C57Bl/6 mice between 8 and 12 weeks old either a purified low-fat (LF; Harlan Teklad TD.00102) diet, or high saturated milk fat (HF; Harlan Teklad TD.97222) diet for four weeks ad libitum. Diets were formulated by Harlan Teklad and compositions[43]. We γ-irradiated gnotobiotic diets and tested them for sterility prior to use. We anesthetized mice using 10 mg/ml ketamine/xylazine followed by exsanguination and cervical dislocation. We snap-froze cecal contents in liquid nitrogen and stored at −80 °C. Three mice each from LF or HF-fed group were used in this study.

**tRNA extraction, purification, and sequencing**. We lysed cecal contents or pelleted bacteria in 400 μL of 0.3 M NaOAc/HOAc, 10 mM EDTA pH 4.8 with an equal volume acetate saturated phenol chloroform pH 4.8. We placed samples in a reciprocating bead beater for 45 s on maximum intensity using FastPrep lysing matrix B. We centrifuged samples at 21,000 rcf for 15 min at 4 °C before re-extraction and ethanol precipitation of total RNA. We purified tRNA by running RNA pellets on a 10% denaturing polyacrylamide gel and cutting type I and type II tRNA bands identified through UV shadowing. We obtained about 2 μg total tRNA from 25 μg total RNA. We eluted tRNA in 200 mM KCl/50 mM KOAc, ethanol precipitated, resuspended in water.

We performed tRNA sequencing as carried out for human tRNA studies[8]. Briefly, starting from about 2 μg total tRNA, we left 1 μg tRNA untreated, and subjected the remaining 1 μg tRNA to demethylase treatment to remove Watson–Crick face methylations on tRNAs known to be present in bacterial tRNAs that can impede the processivity of reverse transcription (such as $m^1A$ or $m^1G$). We performed reverse transcription of 100 ng of demethylase treated and untreated tRNA with a thermophilic reverse transcriptase (TGIRT), which synthesizes cDNA with high processivity. TGIRT only requires base pairing with the 3' terminal nucleotide, which is adenosine in all mature tRNAs, hence no sequence-specific primer is needed. We purified and eluded cDNAs on a 10% polyacrylamide denaturing gel, circularized cDNA and followed by PCR amplification of 12 cycles

for Illumina library construction. We sequenced each tRNA sample twice: with demethylase treatment to generate data for stringent abundance measurement and without demethylase treatment to generate data for modification analysis.

**16s rRNA gene sequencing**. We used a fraction of total cecal contents to extracted genomic DNA by lysing samples in 100 μL water with an equal volume phenol/chloroform and 100 μL of 0.5 mm glass beads. We then performed five 1 min vortexing cycles followed by1 min on ice in between each cycle. We then centrifuged samples for 15 min at 21,000 × *g* at 4 °C, re-extracted, and used ethanol precipitation to yield purified genomic DNA. We then produced amplicon libraries targeting the V4 region of the 16S rRNA gene using barcoded PCR primers 515F (5′-GTGCCAGCMGCCGCGGTAA-3′) and 806R (5′-GGACTACHVGGGTWTC TAAT-3′) for Illumina MiSeq[44]. Sequencing of these libraries at the Environmental Sample Preparation and Sequencing Facility (ESPSF) at Argonne National Laboratory generated 151 bp paired-end amplicon sequences.

**Analysis of the 16S rRNA gene amplicons**. We removed low-quality reads from the raw sequencing results using illumina-utils[45] (available from https://github.com/merenlab/illumina-utils). For the analysis of 16S rRNA gene amplicons data, we used the program 'iu-merge-pairs' with default parameters, which merged partially overlapping paired-end Illumina reads while simultaneously removing any pair with more than 3 mismatches at the overlapped region. For pairs with three or less mismatches, we picked the base to be used in the final merged sequence from the read with the higher Q-score. However, if neither of the reads had a Q-score higher than Q10 at a position of mismatch, we also discarded the pair to diminish the impact of noise to the final results. We used Minimum Entropy Decomposition[29] with default parameters to cluster high-quality 16S rRNA amplicons at 1-nt resolution, and GAST[46] to assign taxonomy to each read individually.

**Analysis of bacterial culture tRNA-seq data**. We performed tRNA-seq analysis for bacterial cultures, each with a single reference genome, in the same way as described in detail for human tRNA-seq analysis[8,20]. A brief description follows. We aligned sequencing reads using Bowtie 1.0[47] to a modified tRNA genome file containing annotated tRNA genes for each of the four bacterial species from Genomic tRNA database[32]. When needed, we appended 3′-CCA to tRNA genes so that all tRNA genes end with 3′-CCA. Prior to mapping, we processed reads using Trimmomatic v0.32[48] using default parameters to remove de-multiplexing and to remove primers, adapters, or any other low-quality sequences. We aligned sequences longer than 15 bp to the library using Bowtie 1.0 with sensitive options using the highest allowed mismatch settings for Bowtie 1.0. We mapped reads to all references simultaneously, and we reported only one alignment declared as valid by the respective mapping software for each read. For each position in the reference file, we calculated the following: at each position, how many total counts were present, how many mutations, and how many aligned reads stopped at the position.

**Analysis of microbiome tRNA-seq data**. Because tRNA molecules (74–96 nucleotides) are shorter than 100 bases, the short insert size results in sequencing of the same molecule twice and the additional sequencing of the Illumina adapters. As a result, the overlapping region in the alignment of the first read and the reverse-complement of the second read in a given pair represents the sequence of the tRNA molecule, and the tailing ends represent the Illumina adapters. To recover tRNA sequences in this dataset we also used the program 'iu-merge-pairs', but with two additional flags and an additional parameter: --retain-only-overlap (to keep only the overlapping region while trimming tailing ends from both reads in a pair), --marker-gene-stringent (to align the reverse-complement of the second read even if the alignment occurs between the end of the second read and the beginning of the first read --which is a unique case for short inserts), and --max-num-mismatches 0 (to remove any pair if there was a disagreement between the aligned portions to increase the quality dramatically as described in ref. [45]). We developed a software tool to identify tRNA sequences in the raw sequencing results: tRNA-seq-tools (available from https://github.com/merenlab/tRNA-seq-tools). We used the tRNA-seq-tools program 'trna-profile' with default parameters to identify all sequences that matched our criteria for standard tRNA (Supplementary Fig. 3). Standard tRNAs contain several secondary structural regions: the acceptor stem, TΨC stem-loop, anticodon stem-loop, D stem-loop, and a short variable loop of 4–5 residues (type I tRNA) or a long variable stem-loop of 13–22 residues (type II tRNA). Conserved sequences at defined positions include the 3'-CCA, a C and GTTC in the T stem-loop, and TXXX(A/G) in the anticodon loop where XXX is the unique identifier for the anticodon residues. All reads in our sequencing method start from the 3'-CCA, so that each read can be assigned to a unique anticodon (Fig. 3a, Supplementary Fig. 3). From the self-contained database of tRNA sequences 'trna-profile' generated for each sample, we recovered tRNA sequences (full-length or non-full-length but with/or without anticodon), and sequences matching to a specific anticodon using the programs 'trna-get-sequences' and 'trna-gen-anticodon-profile'. We used the full-length tRNA sequences to characterize the community composition de novo using Minimum Entropy Decomposition[29] with default parameters. For taxonomy assignment, we generated a database of tRNA sequences tRNAscan-SE (v1.3.1)[49,50] identified from 4,235

gold-standard bacterial genomes (non-endosymbiont genomes with an assembly level of 'chromosome') stored in the Ensembl Genomes 2016 database[31]. We then queried our database of tRNA sequences with GAST (queried in March 2017)[46] to individually assign taxonomy to each high-quality read in our tRNA sequencing results. We generated hierarchical clustering dendrograms with Bray-Curtis distance metric and Ward clustering algorithm to compare sample community profiles for both tRNA sequences and 16S rRNA gene amplicons.

**Analysis of microbiome tRNA modifications.** (a) Derivation of seed sequences from tRNA reads: To identify modified nucleotides in tRNA by read mapping, we first constructed a reference database consisting of seed sequences to which raw tRNA reads were then mapped back. For each FASTA file containing tRNA reads from each demethylase-treated sample, we grouped identical reads and collapsed them into a single sequence and recorded its count. We then clustered the resulting sequences to further reduce redundancy using CD-HIT (v4.6.8)[51] with sequence identity threshold set to 0.98. To reduce the impact of noise, we discarded any cluster with 5 or less reads. For each remaining cluster, we chose a representative read to generate a seed sequence by performing the following procedure. First, to obtain as many high-quality full-length tRNAs as possible, we used tRNAscan-SE (v2.0)[50] to evaluate the longest read in each cluster and kept only sequences with tRNAscan-SE score ≥50 to ensure that the resulting seed sequence had a high likelihood to fold like a tRNA for subsequent processing. The longest read could be truncated due to the RT stops at the 5' end. In >93% of all clusters, the seed sequence derived from the longest read was consistent with the most abundant read (with 100% sequence identity). Second, truncation of 5' ends in some of the representative reads from different clusters may lead to generation of identical seed sequences. After removing these redundant sequences we obtained the reference database from each demethylase-treated sample for subsequent alignment of tRNA reads.

*(b) Mapping of tRNA reads and identification of mutations:* To obtain mutation signatures we ran the alignment program Bowtie[47] on each seed for both libraries with and without demethylase treatment allowing two mismatches. To minimize sequencing artifacts where mutations might be present because of low coverage for a given seed we kept only the alignment results with total coverage of >50 at each nucleotide position. We then generated an output file for each nucleotide position, with the mutation fraction between 0.05–0.98. We used this range because mutation fractions <0.05 had a higher likelihood to be false positives derived from sequencing errors, whereas mutation fractions >0.98 might be derived from a highly abundant but redundant seed sequence that was removed in the last step of seed sequence identification. We further filtered the result by removing those positions where their mutation was only to one of the three possible sequences. These sites likely corresponded to single nucleotide polymorphisms in tRNA sequences. This step was justified because all bacterial culture mutations (Figs 1, 2) had two or three sequence changes, albeit at different ratios. For example, s4U produced both U-to-C and U-to-A mutations, but U-to-C is present at higher frequency than U-to-A. We performed one last filter to remove redundant entries from the seeds that differed by two nucleotides because these generated identical mutation results once we allowed two mismatches in Bowtie alignment.

We analyzed the alignment results by converting to binary BAM format, sorted and indexed using SAMtools (v1.5)[52]. We calculated the overall mutation rate and fractions of mutated nucleotide for each position in each seed sequence and kept only positions with depth of coverage >50 and mutation rate in the range of 0.05–0.98 for subsequent filtering. We defined positions with 2 zero nucleotide fractions as SNPs (single nucleotide polymorphisms) and removed them from the list of mutation entries. We also discarded the redundant mutation entries with the same nucleotide fractions provided that their seed sequences contained the same anticodon and differed by no more than 2 bases. These seed sequences were originated from the same tRNA sequence but were not grouped into one cluster in the previous steps.

(c) Graphing overall mutation rates for seed sequences in each anticodon family: We performed multiple sequence alignments for all seed sequences in each anticodon family by using MAFFT (v7.369b)[53]. For each aligned position we calculated a weighted sum of mutation rates from all seed sequences while excluding gaps and SNPs from the summation.

As expected, the degree of identification of mutation signatures depends on the sequencing depths of the individual sample. As an example, the $m^1A$ and $s^4U$ signatures in *Lactobacillus* and *Bifidobacterium* tRNA$^{Ser}$(UGA) (Fig. 5a) can be found in all three LF samples and two of the three HF samples (HF02 and HF03), consistent with the sequencing coverage of these samples (Supplementary Table 2).

**Analysis of the microbiome metaproteomics data.** In the microbiome metaproteomics study from Mack, Stintzi, Figeys, and co-workers (MetaPro-IQ)[36], 849 proteins were identified that have significantly different expression levels between high-fat and low-fat-fed mouse gut microbiome. We obtained the individual expression level of these proteins from Table S5 of the MetaPro-IQ paper. The MetaPro-IQ authors performed an iterative database search strategy in the MeaPro-IQ workflow to identify the protein sequences using the gut microbial gene catalog from reference[54] as the reference database. We then requested this catalog of mouse gut metagenome comprising ~2.6 million nonredundant genes

from the authors and extracted the 849 coding sequences corresponding to the above 849 proteins according to their entry IDs.

We obtained the taxonomic assignment for each of the 849 proteins using BLASTp, followed by converting the GI# of the top hit to the taxon ID. Among the hits, 98.9% had *e*-value <*e*−14, these were used for subsequent analysis. Among those, 96.8% had >50% amino acid sequence identity and the remaining had an identity between 26.7 and 49.8%. We found that 77% of these (655/849) most closely match or resemble proteins in class Clostridia. These 655 proteins were analyzed further by counting the frequency of the amino acid residues or codons, either singly or in adjacent pairs, and comparing the protein groups that are highly over-expressed or depleted in HF versus LF samples.

**Code availability.** The source code of tRNA-seq-tools is accessible at https://github.com/merenlab/tRNA-seq-tools.

## Data availability
The raw sequencing data have been deposited in NCBI GEO database, accession number GSE100263. Figures 1–6 have associated raw data from the deposited raw sequencing data above. Figure 7 raw data was from the published work in ref. 36. A Reporting Summary for this Article is available as a Supplementary Information file.

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

## Acknowledgements

The authors are grateful to Vanessa Leone for help with the germ free mouse colony, Rich Fox for help with the computer systems administration, and Mitch Sogin for comments on the manuscript. We also thank Drs. Xu Zhang and Daniel Figeys for providing the 849 protein sequences of the mouse gut microbiome metaproteomics study, and Drs. Karsten Kristiansen, Liang Xiao, and Zhongkui Xia for the mouse gut metagenome catalog. Haipeng Wang acknowledges the support from the National Natural Science Foundation of China (31500669), Shandong Provincial Natural Science Foundation (ZR2014FQ024), and China Scholarship Council (201708370053). This work was supported by grants from the NIH P30DK042086 (pilot to T.P. and A.M.E.), P30DK020595 (pilot to T.P.) and RM1HG008935 (to T.P.), the University of Chicago startup funds (to A.M.E.), and the Keck Foundation (to T.P., A.M.E., and E.B.C.).

## Author contributions

M.H.S. performed all tRNA-seq experiments. H.W. performed microbiome tRNA modification analyses. J.N.P. and H.W. performed meta-proteomics analysis. M.J.E., D.W.P., M.P., and J.X. contributed to bioinformatics analyses. W.C.C. and T.P. performed tRNA-seq analysis of bacterial cultures. S.M.O. performed library preparation and sequencing for 16S rRNA gene amplicons. B.L.C., K.M., and E.B.C. provided mouse cecum samples. S.C. and A.M.E. developed the tRNA-seq-tools. M.H.S., T.P., and A.M.E. conceived the project, analyzed data, generated figures, and wrote the paper.

## Additional information

**Competing interests:** The authors declare no competing interests.

