## [Peer Review File · Nature Communications]

Reviewers' comments:

Reviewer #1 (Remarks to the Author):

This is an interesting study that developed new tools for the molecular profiling of microbial transfer RNA, providing potentially important new insights into the functional regulation of microbiota community dynamics. By further developing previously described sequencing technology and cleverly applying new bioinformatics approaches to characterize global microbiome tRNA sequence and modification analysis, a comparative genomics analysis was performed which demonstrated a possible major discrepancy between 16S rDNA amplicon and tRNA based sequencing methods. A detailed work up was presented using gold standard microbial tRNA signatures and then a deep dive was performed to explore class/phyla differences at the gut microbiota community level in a mouse model fed a high fat diet. Although this reviewer feels that this work falls a little short in making taxon-specific claims, potentially important and paradigm shifting observations are made at the class/phylum level. Notably, dietary induced shifts in Clostridia, Proteobacteria and Bacteroides in a complex microbiota community fed a high fat diet shows functional tRNAseq divergence compared with 16S rDNA sequencing. The regulatory mechanism for this disparity is suggested to be at the posttranscriptional level by modifying specific anticodons thereby providing translational stop signals. It is possible that that is could represent an important epigenetic signal in dietary regulation of microbiota community dynamics. This finding alone will likely provide broad appeal to the general readership, but a major critique remains in that this finding is not validated at any level. The authors nicely show the power of tRNA-seq analysis for illustrating the presence and diversity of characterized and unknown tRNA modifications within single bacteria or a complex microbiota community. Some of these modifications have been experimentally validated by conventional chromatography/mass spec methods, but many have not and much data is not interpretable or is missing entirely. In addition, they show genus specific changes in modification sites of tRNAs from *Lactobacillus* and *Bifidobacterium* and the changes in the mutation fraction of tRNAs from specific taxa at the family level. They also highlight that the consequences of tRNA modifications for bacterial physiology merits further investigations, but from this reviewer's standpoint this is key for this paper. At a minimum, the authors should provide metaproteome data to support this translational stop signal at the phyla/class level, especially since many assumptions and exemptions are made to the analysis as currently presented (which the authors correctly highlight). Further, to make a functional claim the authors assume that all the tRNAseq is specifically assigned for protein synthesis. There is emerging metaproteomics data that potentially supports the author's findings, but this should be provided and demonstrated for the dietary high fat findings in order to support their functional claims. Another minor point that the authors should consider to explain and/or rule out taxonomy differences between the two techniques, includes using propidium monoazide to remove dead cells or environmental free DNAs prior to microbial DNA extraction.

Reviewer #2 (Remarks to the Author):

The paper, "Functional genomics of microbiome transfer RNA by high-throughput sequencing and modification analysis" by Schwartz and Wang et al., uses high-throughput tRNA-sequencing methods to investigate tRNA expression and modifications in bacteria to gain better insight into the physiology of microbial populations. The authors first identified tRNA modifications in four cultured bacterial isolates from four species; two with previously mapped modifications as controls and two with unknown modifications for novel discoveries. They identified tRNA modifications that were highly variable across species, thus suggesting that bacterial tRNA modifications warrant further investigation. Next, they applied tRNA-seq methods to study modifications in microbial communities. Because diet has previously been shown to affect gut microbiome structure, they used both 16S rRNA sequencing and tRNA sequencing to characterize cecal samples from mice fed

either a high-fat (HF) or low-fat (LF) diet. Both tRNA and 16S rRNA gene amplicon sequencing data showed distinct clustering of samples based on diet. Furthermore, class-level taxonomy for the samples were qualitatively similar between the tRNA and 16S rRNA, but disagreed quantitatively. The authors suggest that these differences may be due to the fact that tRNA-seq captures mostly active bacteria while 16S methods are less discriminating, or could be artifacts of database selection or sequencing method biases. They also identified modification patterns that differed between taxonomic classes and looked for modifications that had different mutation fractions between HF and LF samples. M¹A modifications, which are associated with enhanced translational functions, were enriched in HF diets, while s⁴U8 modifications, which are a sensor to elicit cellular responses to UV (and may not be important for gut microbes that are not exposed to UV light), had no clear preference for HF versus LF diets.

To the best of my knowledge, this is the first paper to apply tRNA-seq to microbial community samples. The manuscript is well written and will influence thinking in the field. However, several points in the paper need to be addressed before publication.

First, additional assessment should be included in the discussion section. The HF vs. LF diet experiment showed that 16S rRNA and tRNA-seq can both detect diet-induced changes in the gut microbiome. It is good proof-of-concept that UV related tRNA modifications may not be adjusted in response to dietary differences, but the link between the m¹A modifications and diet should be highlighted and there should be more discussion on the potential implications of this finding and what could be done to follow up on it. It would also be helpful to include a paragraph with recommendations on how to best utilize these methods in the current state (highlighting required input amount of RNA, recommended sequencing depth, confidence in depth of taxonomic characterization, etc.).

- For the microbiome analysis of the samples from the mice fed HF and LF diets, it appears that either three mice from each group or three litters of mice from each group were analyzed (Figure 3). Can you please clarify? Were these mice co-housed? Additionally, in the methods (page 23) it says that "age and litter-matched specific pathogen free (SPF) or germ free (GF) male C57Bl/6 mice" were used. Which are being compared in this paper? Are SPF mice being compared to GF mice or are mice only compared within SPF and GF groups? How variable were the individual samples (compared to the aggregated means presented)?

- Calculations of mutation and stop fractions should be explained in the methods in more detail for clarity.

- The methods used to analyze the 16S rRNA sequencing data are reasonable, however more details should be provided on how many reads were obtained (overall and per sample) and if there was any subsampling of reads prior to analysis. Sample read count statistics should also be reported for the tRNA-seq samples. Although page 11 mentions that there were an average of 2.2 million sequences per demethylase treated sample with unambiguous anticodon assignments (and an average of 3.8 million that were not assigned), it would be informative to see the range of sequencing depths across samples and how consistent they are.

- Was any validation done to determine if this sequencing depth is sufficiently capturing microbial community tRNA modifications? For instance, in Figure 5B, you do not see modifications found in *Bacteroidia*- do these bacteria lack these modifications or did you lack sufficient tRNA sequencing depth of *Bacteroidia* (Figure 3C) to identify these modifications?

- 4,235 bacterial genomes from the Ensembl database were used to assign taxonomy to the tRNA sequences. Can you provide more information in the supplement about which taxonomic groups are represented in the database and how it was generated? (i.e. if I were to use tRNA-seq on my samples of interest, could I see if the taxa in my samples are represented in the database? Are your methods sufficient enough that I could use them to build my own database on taxa of

interest?)

- In Figure 3B, there is a dendrogram showing the relationship between samples from mice fed HF and LF diets. The legend notes that sequences from the samples were clustered into groups using Minimum Entropy Decomposition. What metric was used to cluster the samples for the dendrogram? Furthermore, citation 6 in the main manuscript found it difficult to interpret a neighbor-joining phylogenetic tree relating tRNA sequences, but found a tree using UniFrac clustering better reflected organismal phylogeny. Can you comment on what kind of impact the clustering method may have on the tRNA-seq microbiome findings?

- On page 14, bacterial taxon-dependent modifications were identified in the same microbiome sample. Can you add a comment on how often you see similar modifications in the other samples (ie. how many other samples also have the reported modifications in *Lactobacillus* tRNA in Figure 5A?) Is the tree in Figure 5B based on all HF and LF diet microbiome samples (or does separating samples by diet not affect these observed modifications)?

- On page 27, under "Analysis of tRNA modifications" part (a), the first sentence says that a reference database was constructed of seed sequences to which raw tRNA reads were mapped back. How was the reference database constructed and what program and parameters were used to map the raw reads back?

- tRNA-seq was applied to cultured bacterial isolates. The culture growth is described in the methods (page 23) but please provide details on where the isolates came from (Are they the ATCC strains? Were they cultured from gut samples?)

Additional comments:

- The inclusion of the software tool, tRNA-seq-tools, on github is helpful for researchers who may want to perform tRNA-seq on their own samples. On page 38, it mentions that alignment results were analyzed by an "in-house script". Can you either explain in better detail what this script does in the methods, or provide this script with the github code? Furthermore, throughout the methods section, you should provide information on what version of each software program was used to enable future replication of your analysis/results.

- Could you cite a reference for standard tRNA nomenclature (page 5)?

- Figure 2 is a good plot, but very difficult to read (especially the y-axis). Furthermore, the legend is hard to read and varies between parts a, b, and c. In the boxes, what does the green color represent (Is green 0, gray slightly above 0, and then higher values are red? It is difficult to tell). The dots on the left hand side of the graph are helpful.

- In Figure 3, the colors in part A seem to correspond to the colors in part C, but this should be explicitly stated for clarity.

- In Figure 4A, it would be helpful to add arrows above the workflow indicating the commands/tools in the tRNA-seq-tools software corresponding to each step.

Reviewer #3 (Remarks to the Author):

The manuscript by Schwarz et al. describes a very novel approach to analyzing changes in complex microbial populations, here focusing on the dynamics of the gut microbiome. As an alternative to traditional 16S rRNA gene sequencing for quantitative taxonomic analysis, the

authors have applied a tRNA sequencing method to interrogate the spectrum of tRNA sequences, isoacceptor abundances, and modified ribonucleosides as a tool to assess microbiome populations and dynamics. This is a very innovative approach. The technical quality of the work is high, with a rigorous data analysis pipeline based on strict alignment criteria and the power to map several polymerase-disrupting modifications by both mutations and AlkB dealkylation sensitivity. Given the information content of 30-50 tRNA species and a similar number of modified ribonucleosides in the tRNA population, tRNA sequencing could provide more information for microbiome analysis than the traditional 16S rRNA gene sequencing. As indicated by the title and statements in the abstract and introduction, the authors argue that the tRNA sequencing also has advantages over 16S analysis in that it provides an opportunity for studying functional genomics in the microbiome in the form of translational dynamics. Unfortunately, this is the focus of the major weaknesses of the manuscript: the authors did not demonstrate that tRNA-seq analysis offers advantages over 16S rRNA gene sequencing for quantitative taxonomic analysis of the microbiome, nor did they demonstrate how the tRNA-seq data provides insights into translational dynamics or other aspects of functional genomics. This should not detract from the very interesting results in distinguishing high- and low-fat diets by changes in tRNA copy numbers and AlkB-sensitive modification patterns. There are clear quantitative differences apparent in Figure 6. However, distinguishing the diets could likely have been accomplished with less effort by 16S analysis. While this tRNA-seq method is innovative and offers the potential for analysis of complex microbial populations, the results of the studies point to a method with limited utility in distinguishing taxa. Further, there was a lack of higher-level data analysis that would provide quantitative insights into how diet and other factors in the gut affect the microbiome. There are other problems with the manuscript that suggest that Nature Communications may not be the right venue for its publication.

Major issues

- Page 3, third paragraph: The authors state that LC-MS is a low-throughput technology and that only 3 only three bacterial species have been subjected to systematic analyses of tRNA modifications. These statements are factually incorrect and represent poor scholarship.
- Results, "tRNA modifications in bacterial cultures": This section details the tRNA-seq analysis of polymerase stops and mutations in four bacterial species. The authors assign specific ribonucleoside structures to sites affected by mutations arising during reverse transcription. The authors base these structural identities on the conservation among prokaryotes of specific modifications at specific sites in specific tRNA isoacceptors. However, it is unrigorous and inappropriate to draw structural conclusions from mutation data – structural conclusions must be validated chemically or biochemically. The authors do not provide any form of validation of site-specific modification structure. Further, the authors subsequently show that conclusions based on conservation are incorrect when comparing diverse species. For both these reasons, the authors need to soften their structural conclusions, using statements such as "m1A-like" or "presumptive" to avoid promoting the misconception that mutations can be interpreted as specific structures. This is more than nit-picking and is a major problem with the manuscript.
- This overstatement of modification structures based on mutational signatures is most apparent in two sections. First in the last sentence on page 6, in which the authors conclude that they had identified "all known Watson-Crick face base modifications in these four bacterial species..." A more rigorous statement would have been "mutational signatures consistent with all known...." The second example is line 4 on page 8: "We identified 13 and 12 tRNAs with m1A22 modifications in..." m1A at position 22 was never proved. It is also unclear what the authors mean by "only four have been previously mapped.." Four what? tRNA species? A literature citation is needed here.
- Figure 2 was entirely unreadable, so it is not possible to judge the quality of the mutational analysis.
- Page 12: The only comparison to 16S rRNA sequencing occurred with the tRNA sequence analysis. The authors concluded that the tRNA-based taxonomy "qualitatively matched" the 16S

analysis. This soft conclusion typifies one of the central weaknesses of the work: a lack of quantitative rigor in comparing datasets. Which analysis provided the greatest depth of coverage of the gut microbiome population? The authors point out that the tRNA-seq and 16S analyses showed differing proportions of several bacterial families, which remains a problem without some effort to perform control studies to calibrate the method.

- A major problem with regard to performance relative to 16S analysis is the limitation of the tRNA-seq analysis to the level of bacterial families (Figure 6 and page 17). This was disappointingly poor resolution and it is unclear why higher resolution analyses were not attempted – see the comments below about the lack of rigor in the analysis of datasets.
- Page 18-19: To their credit, the authors were very thoughtful and open in their discussion of the basis for the differences between tRNA-seq and 16S analysis. Both methods were noted to suffer from limitations and weaknesses. However, the authors argument that tRNAs arise mainly from living bacteria is incorrect. tRNA is among the most stable of RNA structures, with tRNAs surviving for days in cell culture medium (see Obregon-Henao et al. PLoS One 7: e29970, 2012). The tRNA species extracted from the fecal material represent free tRNA from dead cells or secreted by living cells, in addition to tRNA from within living cells.
- One of the major shortcomings of the manuscript is the lack of higher level data integration in some form of predictive model based on the high- and low-fat diet comparisons. The authors have an opportunity to perform a true functional genomics analysis with the information-rich tRNA sequence, mutation and modification datasets. Yet the data analysis is limited to single comparisons of a few bacterial families, as in Figure 6. There are clear differences that could form the basis for a predictive model. It was disappointing that there was no effort at functional genomics given the title and statements in the abstract and introduction.

Minor issues

- Figure 1: the red tracing in panel B completely masks the black tracing, which was confusing until magnifying the image revealed the presence of the black line. Is there some way to make this exact overlay clear to the reader?

Point-by-point response

(All significant changes in response to reviewer's comments are shown in red in the text.)

Editorial

In particular, the reviewers (especially #1 and #3) feel that some aspects of the study are preliminary in the absence of further evidence supporting that tRNA-seq analysis offers advantages over 16S rRNA gene sequencing for quantitative taxonomic analysis of the microbiome, or for providing insights into translational dynamics or other aspects of functional genomics.

Response: We have added new analysis to reveal (i) additional insight of tRNA anticodon-based taxonomy; (ii) relationship of m¹A tRNA modification and microbiome protein expression.

(i) Taxonomy (new Figure S7, description on p. 13, third paragraph and p.14): A new avenue of tRNA-seq is the ability to analyze taxonomy based on the 30-45 different anticodons that are present in a bacterial genome instead of a single 16S rRNA sequence. We performed additional analysis of the tRNA^{Glu}(CUC) and tRNA^{Glu}(UUC) results to gain further insight into tRNA anticodon based taxonomy. tRNA^{Glu}(UUC) can read both GAA/GAG codons and is essential in all cells, whereas tRNA^{Glu}(CUC) can only read GAG codon and is optional for life. We found that the occurrence of TTC genes varied widely among the six major bacterial classes identified by both tRNA-seq and 16S-seq (panel a). Zooming into the class Bacilli we found that 75% of all CTC genes (78/104) are concentrated in the *Lactobacillaceae* family (panel b), suggesting that this family can be uniquely assessed using tRNA^{Glu}(CUC) reads. *Lactobacillaceae* can be

resolved both in tRNA-seq and 16S rRNA gene amplicons, showing similar differences between HF and LF microbiome samples (panel c using all tRNA anticodon reads).

Focusing further on tRNA^{Glu}(CUC) and tRNA^{Glu}(UUC) reads from the family of Lactobacillaceae, we found that the ratio of tRNA^{Glu}(CUC) to tRNA^{Glu}(UUC) was higher in the LF samples than the HF samples (panel d). This could be due to either higher tRNA^{Glu}(CUC) gene expression or a higher proportion of Lactobacillaceae organisms that contain tRNA^{Glu}(CUC) in the LF samples. In the case of the former, the increased tRNA^{Glu}(CUC) expression could potentially add an additional capacity for the Lactobacillaceae in LF samples to specifically decode the GAG codon in translation. Finally, we found that all tRNA^{Glu}(CUC) reads in our samples are from the *Lactobacillus* genus in the Lactobacillaceae family. This genus is represented by 31 genomes in our tRNA database; 18 different tRNA^{Glu}(CUC) gene sequences are present among the 31 genomes, which allowed us to investigate the species-level distribution of reads that resolved to tRNA^{Glu}(CUC) (panel e).

Overall, these results indicate that anticodon-level taxonomic analyses of tRNA transcripts can provide additional insights into physiological differences between individual branches of bacteria.

(ii) Proteomics and tRNA modification (new Fig. 7, description on p. 19, third paragraph, and p. 20-22): As discussed in several reviews, proteomics on microbiome is extremely challenging. We were fortunate to find one published work (Zhang et al. *Microbiome* (2016) 4:31, MetaPro-IQ: a universal metaproteomic approach to studying human and mouse gut microbiota) where

metaproteomics by mass spectrometry were performed on mice fed with high-fat or low-fat diet. In that work, a total of 849 proteins were found to have significantly different expression levels between the HF and LF samples collected over a course of 43 days. The day 43 experimental condition most closely resembles the experimental condition of our tRNA-seq experiment. The metaproteomics data from day 43 show a marked difference between HF and LF conditions (panel a). Our taxonomic analysis assigned 77% (655/849) of these proteins to the class Clostridia (panel b), and we focused our subsequent analysis specifically on this group (panel c). To examine a possible relationship between these differentially expressed proteins and tRNA m¹A modifications, we compared the amino acid and codon compositions of these proteins that are highly expressed in HF samples (log>1) and depleted in HF samples (log<-1). In Clostridia, we know that m¹A modification is present in tRNAs that decode amino acids Cys, Glu, Gln, and Ser (Fig. 6a). We combined amino acid residues or codons based on protein expression levels in HF, and subtracted the fractions of HF over-expressed proteins from HF under-expressed proteins (panels d, e). In both cases, Cys and Glu, but not Gln and Ser are over-represented among the highly over-expressed proteins, although this over-representation was not unique to Cys and Glu.

Since ribosome decoding includes an adjacent pair of mRNA codons in the A and P site, we also combined amino acid and codon pairs in a similar fashion (panels f, g). Excitingly, the top five most over-represented amino acid pairs in the HF overexpressed proteins relative to the HF depleted proteins were E-E (Glu-Glu), K-E (Lys-Glu), L-E (Leu-Glu), E-K (Glu-Lys), and R-E (Arg-Glu). For codon pairs, the top five most over-represented were GAG-CAA (Glu-Gln), AAG-CAA (Lys-Gln), GGC-CAA (Gly-Gln), GAG-TCT (Glu-Ser), and ACC-CAA (Thr-Gln). Decoding the top five over-represented amino acid and codon pairs all involves one or two tRNAs containing m¹A modification, in this case tRNA^{Glu} (anticodon UUC, CUC) and tRNA^{Gln} (anticodon UUG, CUG) in HF samples. These proteomic results are therefore consistent with our hypothesis that tRNAs with higher m¹A levels enhance the decoding of their respective amino acid/codon pairs.

Overall, this analysis indicates that tRNA m¹A modification level differences are consistent with m¹A modified tRNAs facilitating increased expression of microbiome proteins with specific amino acid and codon contexts.

A note on title change:

We changed the title based on reviewer's comments. We realized that "functional genomics" may be subject to interpretation of what it means. We changed the title to "Microbiome characterization by high throughput transfer RNA sequencing and modification analysis" to better reflect the nature of our study.

Reviewer #1

This is an interesting study that developed new tools for the molecular profiling of microbial transfer RNA, providing potentially important new insights into the functional regulation of microbiota community dynamics. By further developing previously described sequencing technology and cleverly applying new bioinformatics approaches to characterize global microbiome tRNA sequence and modification analysis, a comparative genomics analysis was performed which demonstrated a possible major discrepancy between 16S rDNA amplicon and tRNA based sequencing methods. A detailed work up was presented using gold standard

microbial tRNA signatures and then a deep dive was performed to explore class/phyla differences at the gut microbiota community level in a mouse model fed a high fat diet. Although this reviewer feels that this work falls a little short in making taxon-specific claims, potentially important and paradigm shifting observations are made at the class/phylum level. Notably, dietary induced shifts in Clostridia, Proteobacteria and Bacteroides in a complex microbiota community fed a high fat diet shows functional tRNAseq divergence compared with 16S rDNA sequencing. The regulatory mechanism for this disparity is suggested to be at the posttranscriptional level by modifying specific anticodons thereby providing translational stop signals. It is possible that that is could represent an important epigenetic signal in dietary regulation of microbiota community dynamics. This finding alone will likely provide broad appeal to the general readership....

Response: We thank the reviewer for the enthusiastic and encouraging comments.

...but a major critique remains in that this finding is not validated at any level. The authors nicely show the power of tRNA-seq analysis for illustrating the presence and diversity of characterized and unknown tRNA modifications within single bacteria or a complex microbiota community. Some of these modifications have been experimentally validated by conventional chromatography/mass spec methods, but many have not and much data is not interpretable or is missing entirely. In addition, they show genus specific changes in modification sites of tRNAs from *Lactobacillus* and *Bifidobacterium* and the changes in the mutation fraction of tRNAs from specific taxa at the family level. They also highlight that the consequences of tRNA modifications for bacterial physiology merits further investigations, but from this reviewer's standpoint this is key for this paper. At a minimum, the authors should provide metaproteome data to support this translational stop signal at the phyla/class level, especially since many assumptions and exemptions are made to the analysis as currently presented (which the authors correctly highlight). Further, to make a functional claim the authors assume that all the tRNAseq is specifically assigned for protein synthesis. There is emerging metaproteomics data that potentially supports the author's findings, but this should be provided and demonstrated for the dietary high fat findings in order to support their functional claims.

Response: We addressed two different points raised here:

(i) The m¹A modification fraction differences do not represent translation stops, rather, they are proposed to enhance ribosome utilization of the modified tRNAs. At the most basic level, our hypothesis is that m¹A hypermodified tRNAs more benefit the translation of selected proteins. This idea was supported by a human m¹A-modified tRNA study where the m¹A-modified tRNA has an increased affinity for the EF-1A protein (equivalent to bacterial EF-Tu) that delivers tRNA to the ribosome, and increased the expression of a reporter gene (Liu et al., *Cell* (2016) 167, 816-828. ALKBH1-Mediated tRNA Demethylation Regulates Translation). This clarification is now added on p. 19, first paragraph.

(ii) To get to the main point of relating our tRNA modification results to microbiome metaproteomics: as discussed in the literature, proteomics on microbiome is extremely challenging. We were fortunate to find one published work (Zhang et al. *Microbiome* (2016) 4:31, MetaPro-IQ: a universal metaproteomic approach to studying human and mouse gut microbiota) where metaproteomics by mass spectrometry were performed on mice fed with high-fat or low-fat diet. We analyzed these published data and established that our results are

consistent with higher m¹A modified tRNA enhancing translation of differentially expressed proteins between HF and LF-fed mouse microbiome. The data analysis and interpretation are described under Editorial comment (ii), and in the text and figure on p. 19-22.

Another minor point that the authors should consider to explain and/or rule out taxonomy differences between the two techniques, includes using propidium monoazide to remove dead cells or environmental free DNAs prior to microbial DNA extraction.

Response: Reviewer 3 actually suggests that our technique may not be able to distinguish live and dead bacteria, a point we simply suggested in the Discussion section. We have deleted this suggestion of tRNA-seq regarding live or dead bacteria.

Reviewer #2

The paper, "Functional genomics of microbiome transfer RNA by high-throughput sequencing and modification analysis" by Schwartz and Wang et al., uses high-throughput tRNA-sequencing methods to investigate tRNA expression and modifications in bacteria to gain better insight into the physiology of microbial populations. The authors first identified tRNA modifications in four cultured bacterial isolates from four species; two with previously mapped modifications as controls and two with unknown modifications for novel discoveries. They identified tRNA modifications that were highly variable across species, thus suggesting that bacterial tRNA modifications warrant further investigation. Next, they applied tRNA-seq methods to study modifications in microbial communities. Because diet has previously been shown to affect gut microbiome structure, they used both 16S rRNA sequencing and tRNA sequencing to characterize cecal samples from mice fed either a high-fat (HF) or low-fat (LF) diet. Both tRNA and 16S rRNA gene amplicon sequencing data showed distinct clustering of samples based on diet. Furthermore, class-level taxonomy for the samples were qualitatively similar between the tRNA and 16S rRNA, but disagreed quantitatively. The authors suggest that these differences may be due to the fact that tRNA-seq captures mostly active bacteria while 16S methods are less discriminating, or could be artifacts of database selection or sequencing method biases. They also identified modification patterns that differed between taxonomic classes and looked for modifications that had different mutation fractions between HF and LF samples. M¹A modifications, which are associated with enhanced translational functions, were enriched in HF diets, while s⁴U8 modifications, which are a sensor to elicit cellular responses to UV (and may not be important for gut microbes that are not exposed to UV light), had no clear preference for HF versus LF diets.

To the best of my knowledge, this is the first paper to apply tRNA-seq to microbial community samples. The manuscript is well written and will influence thinking in the field. However, several points in the paper need to be addressed before publication.

Response: We thank the reviewer for the enthusiastic and encouraging comments.

(1) First, additional assessment should be included in the discussion section. The HF vs. LF diet experiment showed that 16S rRNA and tRNA-seq can both detect diet-induced changes in the gut microbiome. It is good proof-of-concept that UV related tRNA modifications may not be adjusted in response to dietary differences, but the link between the m¹A modifications and diet

should be highlighted and there should be more discussion on the potential implications of this finding and what could be done to follow up on it. It would also be helpful to include a paragraph with recommendations on how to best utilize these methods in the current state (highlighting required input amount of RNA, recommended sequencing depth, confidence in depth of taxonomic characterization, etc.).

Response: Following up the comments by reviewers 1 and 3, we analyzed a published microbiome metaproteomic data under the same condition as our tRNA-seq samples. These new results are described under Editorial comment (ii), and in the text and figure on p. 19-22.

We also added a description with recommendations on how to best utilized tRNA-seq in the current state (p. 23, second paragraph).

(2) - For the microbiome analysis of the samples from the mice fed HF and LF diets, it appears that either three mice from each group or three litters of mice from each group were analyzed (Figure 3). Can you please clarify? Were these mice co-housed? Additionally, in the methods (page 23) it says that “age and litter-matched specific pathogen free (SPF) or germ free (GF) male C57Bl/6 mice” were used. Which are being compared in this paper? Are SPF mice being compared to GF mice or are mice only compared within SPF and GF groups? How variable were the individual samples (compared to the aggregated means presented)?

Response: Three mice each from the HF or LF-fed group were analyzed. We also clarified the information of SPF mice and the diets in the Method section (p. 30, second paragraph).

(3) - Calculations of mutation and stop fractions should be explained in the methods in more detail for clarity.

Response: tRNA-seq analysis for bacterial cultures, each with a single reference genome, is the same as that used for human tRNA-seq analysis already published. We added a brief description on p. 32, last paragraph.

We also added a new supplemental figure (Fig. S8) that shows a step-by-step flow diagram of the microbiome tRNA modification analysis.

(4) - The methods used to analyze the 16S rRNA sequencing data are reasonable, however more details should be provided on how many reads were obtained (overall and per sample) and if there was any subsampling of reads prior to analysis. Sample read count statistics should also be reported for the tRNA-seq samples. Although page 11 mentions that there were an average of 2.2 million sequences per demethylase treated sample with unambiguous anticodon assignments (and an average of 3.8 million that were not assigned), it would be informative to see the range of sequencing depths across samples and how consistent they are.

Response: We added these information in supplemental Table S2.

(5) - Was any validation done to determine if this sequencing depth is sufficiently capturing microbial community tRNA modifications? For instance, in Figure 5B, you do not see modifications found in *Bacteroidia*- do these bacteria lack these modifications or did you lack

sufficient tRNA sequencing depth of *Bacteroidia* (Figure 3C) to identify these modifications?

Response: We added a brief description on which bacterial families we have identified to contain m¹A and s⁴U modifications (p. 37, second paragraph). Our culture study of *B. viscericola* which belong to class Bacteroidetes (Fig. 2) did not find any m¹A and s⁴U modification; we therefore do not expect to find these modifications in our microbiome samples. This point is clarified on p. 17, last paragraph.

(6) - 4,235 bacterial genomes from the Ensembl database were used to assign taxonomy to the tRNA sequences. Can you provide more information in the supplement about which taxonomic groups are represented in the database and how it was generated? (i.e. if I were to use tRNA-seq on my samples of interest, could I see if the taxa in my samples are represented in the database? Are your methods sufficient enough that I could use them to build my own database on taxa of interest?)

Response: The bacterial genomes in each class we used for tRNA taxonomy analysis in this work are included in Table S4.

(7) - In Figure 3B, there is a dendrogram showing the relationship between samples from mice fed HF and LF diets. The legend notes that sequences from the samples were clustered into groups using Minimum Entropy Decomposition. What metric was used to cluster the samples for the dendrogram? Furthermore, citation 6 in the main manuscript found it difficult to interpret a neighbor-joining phylogenetic tree relating tRNA sequences, but found a tree using UniFrac clustering better reflected organismal phylogeny. Can you comment on what kind of impact the clustering method may have on the tRNA-seq microbiome findings?

Response: We thank the reviewer for pointing this out. We now have added a description regarding the generation of the dendrograms in the Method section (p. 34, last paragraph).

These dendrograms have no phylogenetic meaning, and they simply represent the results of hierarchical clustering of samples based on their 16S rRNA gene amplicon or tRNA sequence profiles. We did survey multiple different ways to cluster our data using different widely used distance metrics and we did not observe a significant change regarding how samples related to one another. We elected to use Bray-Curtis distance estimate, which is one of the most common distance metrics used in similar ecological studies. While the structures of our dendrograms were preserved across different clustering approaches, this does not necessarily mean different algorithms to infer sequence profiles in samples would also have resulted in identical organization of samples. Although we did not investigate this, Minimum Entropy Decomposition (MED) is one of the most reliable algorithms to infer community structures through sequencing data, and it would be unlikely to see major differences across comparable widely used algorithms.

(8) - On page 14, bacterial taxon-dependent modifications were identified in the same microbiome sample. Can you add a comment on how often you see similar modifications in the other samples (ie. how many other samples also have the reported modifications in *Lactobacillus*

tRNA in Figure 5A?) Is the tree in Figure 5B based on all HF and LF diet microbiome samples (or does separating samples by diet not affect these observed modifications)?

Response: We found the same *Lactobacillus* m¹A modification (Fig. 5A) for 2/3 HF samples and 3/3 LF samples. The only one missing is for HF-01 because of its low sequencing coverage. We added a brief description on this on p. 37, second paragraph.

(9) - On page 27, under “Analysis of tRNA modifications” part (a), the first sentence says that a reference database was constructed of seed sequences to which raw tRNA reads were mapped back. How was the reference database constructed and what program and parameters were used to map the raw reads back?

Response: These have been described in detail in the Method section (p. 35, first paragraph) and in Fig. S8 for more details. We also add the total number of seed sequences obtained from each demethylase-treated sample in Table S2.

(10) - tRNA-seq was applied to cultured bacterial isolates. The culture growth is described in the methods (page 23) but please provide details on where the isolates came from (Are they the ATCC strains? Were they cultured from gut samples?)

Response: These have been added on p. 30, first paragraph.

Additional comments:

(1) - The inclusion of the software tool, tRNA-seq-tools, on github is helpful for researchers who may want to perform tRNA-seq on their own samples. On page 38, it mentions that alignment results were analyzed by an “in-house script”. Can you either explain in better detail what this script does in the methods, or provide this script with the github code? Furthermore, throughout the methods section, you should provide information on what version of each software program was used to enable future replication of your analysis/results.

Response: We rewrote this sentence for better clarification.

(2) - Could you cite a reference for standard tRNA nomenclature (page 5)?

Response: Reference added (22 in main text). These nomenclatures were based on the yeast tRNA^{Phe} crystal structures solved in the 1970s.

(3) - Figure 2 is a good plot, but very difficult to read (especially the y-axis). Furthermore, the legend is hard to read and varies between parts a, b, and c. In the boxes, what does the green color represent (Is green 0, gray slightly above 0, and then higher values are red? It is difficult to tell). The dots on the left hand side of the graph are helpful.

Response: We remade Fig. 2 for better visualization and explanation. The same panels have been separated into a new Fig. 2 and supplemental Fig. S3, and Fig. 2a labels have been improved.

(4) - In Figure 3, the colors in part A seem to correspond to the colors in part C, but this should be explicitly stated for clarity.

Response: Done.

(5) - In Figure 4A, it would be helpful to add arrows above the workflow indicating the commands/tools in the tRNA-seq-tools software corresponding to each step.

Response: Added.

Reviewer #3

The manuscript by Schwarz et al. describes a very novel approach to analyzing changes in complex microbial populations, here focusing on the dynamics of the gut microbiome. As an alternative to traditional 16S rRNA gene sequencing for quantitative taxonomic analysis, the authors have applied a tRNA sequencing method to interrogate the spectrum of tRNA sequences, isoacceptor abundances, and modified ribonucleosides as a tool to assess microbiome populations and dynamics. This is a very innovative approach. The technical quality of the work is high, with a rigorous data analysis pipeline based on strict alignment criteria and the power to map several polymerase-disrupting modifications by both mutations and AlkB dealkylation sensitivity. Given the information content of 30-50 tRNA species and a similar number of modified ribonucleosides in the tRNA population, tRNA sequencing could provide more information for microbiome analysis than the traditional 16S rRNA gene sequencing. As indicated by the title and statements in the abstract and introduction, the authors argue that the tRNA sequencing also has advantages over 16S analysis in that it provides an opportunity for studying functional genomics in the microbiome in the form of translational dynamics.

Response: We thank the reviewer for the enthusiastic and encouraging comments.

(1) Unfortunately, this is the focus of the major weaknesses of the manuscript: the authors did not demonstrate that tRNA-seq analysis offers advantages over 16S rRNA gene sequencing for quantitative taxonomic analysis of the microbiome, nor did they demonstrate how the tRNA-seq data provides insights into translational dynamics or other aspects of functional genomics. This should not detract from the very interesting results in distinguishing high- and low-fat diets by changes in tRNA copy numbers and AlkB-sensitive modification patterns. There are clear quantitative differences apparent in Figure 6. However, distinguishing the diets could likely have been accomplished with less effort by 16S analysis. While this tRNA-seq method is innovative and offers the potential for analysis of complex microbial populations, the results of the studies point to a method with limited utility in distinguishing taxa. Further, there was a lack of higher-level data analysis that would provide quantitative insights into how diet and other factors in the gut affect the microbiome.

Response: As suggested by reviewers 3 and 1 and the editors, we performed additional analysis on (i) the type of new information that can be learned by tRNA-seq using anticodon-based

taxonomy and (ii) a relationship of microbiome metaproteomics and tRNA m¹A modification. Our results are consistent with tRNA-seq providing new insights relative to 16S-seq, and protein expression is enhanced upon higher tRNA m¹A levels in specific contexts. These are described under editorial comments (i) and (ii) and added in the main text (p.13-14,10-22) and new figures (Fig. S6 for anticodon-taxonomy, Fig. 7 for m¹A and metaproteome protein expression).

Major issues

(1)• Page 3, third paragraph: The authors state that LC-MS is a low-throughput technology and that only 3 only three bacterial species have been subjected to systematic analyses of tRNA modifications. These statements are factually incorrect and represent poor scholarship.

Response: We do not mean disrespect to LC-MS, it is an extremely powerful technique for RNA modification studies. As a matter of fact, we have done and still do LC-MS on tRNA modifications in our own research.

We want to thank the reviewer for pointing out that our initial assessment on systematic analyses of bacterial cultures was incorrect. We have now found five bacterial species whose tRNA modifications have been “systematically” determined, defined as >2/3rd of all tRNA species were mapped.

We have revised the text and references accordingly to amend these descriptions (p. 3, last paragraph).

(2)• Results, “tRNA modifications in bacterial cultures”: This section details the tRNA-seq analysis of polymerase stops and mutations in four bacterial species. The authors assign specific ribonucleoside structures to sites affected by mutations arising during reverse transcription. The authors base these structural identities on the conservation among prokaryotes of specific modifications at specific sites in specific tRNA isoacceptors. However, it is unrigorous and inappropriate to draw structural conclusions from mutation data – structural conclusions must be validated chemically or biochemically. The authors do not provide any form of validation of site-specific modification structure. Further, the authors subsequently show that conclusions based on conservation are incorrect when comparing diverse species. For both these reasons, the authors need to soften their structural conclusions, using statements such as “m¹A-like” or “presumptive” to avoid promoting the misconception that mutations can be interpreted as specific structures. This is more than nit-picking and is a major problem with the manuscript.

Response: We agree that tRNA-seq can only infer the chemical structure, but not conclusively determine the chemical structure without additional information. We have revised the text throughout to amend these descriptions (e.g. p. 6 paragraph).

(3)• This overstatement of modification structures based on mutational signatures is most apparent in two sections. First in the last sentence on page 6, in which the authors conclude that they had identified “all known Watson-Crick face base modifications in these four bacterial species...” A more rigorous statement would have been “mutational signatures consistent with all known...”

The second example is line 4 on page 8: “We identified 13 and 12 tRNAs with m¹A22 modifications in...” m¹A at position 22 was never proved. It is also unclear what the authors

mean by “only four have been previously mapped..” Four what? tRNA species? A literature citation is needed here.

Response: We have revised the text accordingly to amend these descriptions (e.g. p. 8, first paragraph), including citing appropriate literature on *B. subtilis* m¹A modifications.

(4)• Figure 2 was entirely unreadable, so it is not possible to judge the quality of the mutational analysis.

Response: We revised Fig. 2 for better visualization and explanation. The same panels have been separated into a new Fig. 2 and supplemental Fig. S3, and Fig. 2a labels have been improved.

(5)• Page 12: The only comparison to 16S rRNA sequencing occurred with the tRNA sequence analysis. The authors concluded that the tRNA-based taxonomy “qualitatively matched” the 16S analysis. This soft conclusion typifies one of the central weaknesses of the work: a lack of quantitative rigor in comparing datasets. Which analysis provided the greatest depth of coverage of the gut microbiome population? The authors point out that the tRNA-seq and 16S analyses showed differing proportions of several bacterial families, which remains a problem without some effort to perform control studies to calibrate the method.

Response: This is similar to the main editorial comment (i, anticodon-based taxonomy) regarding the comparison of tRNA-seq and 16S-seq. We performed additional analysis, added a new Fig. S7, and description in the text (p. 13, third paragraph and p.14).

For more detailed description please refer to the detailed description under editorial point (i).

(6)• A major problem with regard to performance relative to 16S analysis is the limitation of the tRNA-seq analysis to the level of bacterial families (Figure 6 and page 17). This was disappointingly poor resolution and it is unclear why higher resolution analyses were not attempted – see the comments below about the lack of rigor in the analysis of datasets.

Response: This is again similar to the main editorial (i, anticodon-based taxonomy) regarding the comparison of tRNA-seq and 16S-seq. We performed additional analysis, added a new Fig. S7, and description in the text (p. 13, third paragraph and p.14).

For more detailed description please refer to the detailed description under editorial point (i).

(7)• Page 18-19: To their credit, the authors were very thoughtful and open in their discussion of the basis for the differences between tRNA-seq and 16S analysis. Both methods were noted to suffer from limitations and weaknesses. However, the authors argument that tRNAs arise mainly from living bacteria is incorrect. tRNA is among the most stable of RNA structures, with tRNAs surviving for days in cell culture medium (see Obregon-Henao et al. PLoS One 7: e29970, 2012). The tRNA species extracted from the fecal material represent free tRNA from dead cells or secreted by living cells, in addition to tRNA from within living cells.

Response: The point of tRNAs mainly arose from living bacteria was a suggestion in the Discussion section. Thanks for pointing out our mistake. We have deleted this suggestion in the text.

(8)• One of the major shortcomings of the manuscript is the lack of higher level data integration in some form of predictive model based on the high- and low-fat diet comparisons. The authors have an opportunity to perform a true functional genomics analysis with the information-rich tRNA sequence, mutation and modification datasets. Yet the data analysis is limited to single comparisons of a few bacterial families, as in Figure 6. There are clear differences that could form the basis for a predictive model. It was disappointing that there was no effort at functional genomics given the title and statements in the abstract and introduction.

Response: As described under editorial comments (i) and (ii), we now include additional analyses to elaborate on anticodon-based taxonomy and tRNA expression, as well as relate our tRNA modification result with microbiome metaproteomics.

That said, we realized that “functional genomics” may be subject to interpretation of what it means. We changed the title to “Microbiome characterization by high throughput transfer RNA sequencing and modification analysis” to better reflect the nature of our study.

Minor issues

(1)• Figure 1: the red tracing in panel B completely masks the black tracing, which was confusing until magnifying the image revealed the presence of the black line. Is there some way to make this exact overlay clear to the reader?

Response: The baselines of Figs. 1 and 4 panels have been raised which clearly visualize the presence of the lines with zero values.

REVIEWERS' COMMENTS:

Reviewer #1 (Remarks to the Author):

The authors have responded to reviewer critique in exemplary fashion and have generated a highly impactful study.

Reviewer #2 (Remarks to the Author):

The submitted manuscript has substantially improved through the first round of revisions and should be accepted for publication with a few additions. Below are previously identified concerns, how they were addressed by the authors, and additional comments.

(1) The original paper did not provide sufficient evidence for the translational or functional insights that tRNA-seq analyses may provide over existing 16S rRNA seq methods. The updated manuscript includes new analysis on (a) tRNA anticodon-based taxonomy and (b) the relationship of a tRNA modification with microbiome protein expression that better highlight the utility of the new tool.

a. tRNA anticodon-based taxonomy analysis: Analysis of tRNA^{Glu} (CUC) and tRNA^{Glu} (UUC) revealed that the occurrence of TTC genes varied among bacterial classes and, in particular, can be used to distinguish *Lactobacillaceae* in mice fed HF and LF diets, and even allowed for species level analysis of these bacteria.

b. Proteomics and tRNA modification analysis: The authors integrated analysis from a previously published proteomics study looking at mice fed HF and LF diets, where they performed mass spectrometry and identified proteins with significantly different expression between the two diet groups. It is impressive that the authors were able to validate some of their findings from the tRNA-seq experiment with metabolomics data from an independent study. I think there needs to be more discussion on the comparisons being made (see below), but otherwise think that it is a convincing example of how tRNA-seq and metaproteomics could be utilized in future studies.

(2) The authors clarified appropriate details regarding sample sizes and collection, sequencing depth, reference database composition, and metric calculations. While the sample sizes for used (3 HF, 3 LF) were small, they are reconfirming microbiome differences from previous work and using it as a pilot to show that their method is feasible and has important implications. Figures were also sufficiently updated for clarity. Some additional points that remain to be clarified include:

a. Page 21- "The day 43 experimental condition most closely resembles the experimental condition of our tRNA-seq experiment"- Why? What were the major differences in experimental conditions (if any) that could impact the findings? In the tRNA-seq experiment, mice were fed diets for four weeks (28 days) before sample collection—in the MetaPro-IQ approach, mice were fed a HF or LF diet for 43 days, and sampled on days 0, 14, 29, and 43—please clarify why didn't you use the day 29 data.

b. Page 33- You mention Trimmomatic v0.32 was used to "remove primers, adapters, or any other low-quality sequences." Please clarify the cutoff for a low-quality sequence (did you use default parameters or specify something different?)

c. Page 37- "We obtained the taxonomic assignment for each of the 849 proteins using BLASTp"- What parameters (percent identity, evalue, etc) were used?

* Although minor additions, these details are especially important for clarity and reproducibility.

(3) Sufficient information was added to the discussion discussing the limitations and best-use cases of the methods described. I also think that the name change of the article better reflects the contents of the paper.

Reviewer #3 (Remarks to the Author):

In this revised manuscript, the authors have responded to reviewer comments with clarifications and new data from a gut microbiome proteomics publication. While the manuscript has been significantly improved in terms of clarity, accuracy, and scholarship, the central argument that tRNA sequencing offers advantages for microbiome analysis than the traditional 16S rRNA gene sequencing is still not proven. The authors still have not demonstrated that tRNA-seq analysis offers advantages over 16S rRNA gene sequencing for quantitative taxonomic analysis of the microbiome or for identifying bacteria above the level of genus.

The authors have now expanded the manuscript to include a published gut microbiome metaproteomics analysis of the changes in the microbiome caused by high- and low-fat diets. The proteomics data was analysed in terms of the frequencies of amino acid usage in proteins differentially regulated by the diet. They coordinated this amino acid data with codon frequencies arising from the tRNA-seq analysis. While this is a very interesting analysis of microbiome proteomics and genomics, the proteomics data from another mouse study performed at a different institution two years ago cannot be compared to tRNA-seq data from the present study. Even though the same mouse background was used in both studies (C57BL/6), it is now well established in the literature that gut microbiomes differ dramatically from institution to institution, and within an institution at different times. Even with mice purchased from the same vendor at the same time but housed in different cages will have different microbiomes. Further, a different mouse chow was used for the high- and low-fat diet in the two studies. It is simply not credible to compare the published proteomics data to the present tRNA-seq data. If the proteomics data had been obtained from the animals used to perform tRNA-seq analysis, then these would be extremely exciting studies.

The results showing how tRNA sequences and tRNA modifications distinguish high- and low-fat diets are clear and novel. However, given that other sequencing-based approaches, including 16S analysis and metagenomics, could distinguish the dietary states, it is not clear that the manuscript rises to the level of novelty and impact expected for Nature Communications.

NCOMMS-18-14165B: Microbiome characterization by high-throughput transfer RNA sequencing and modification analysis

REVIEWERS' COMMENTS:

Reviewer #1 (Remarks to the Author):

The authors have responded to reviewer critique in exemplary fashion and have generated a highly impactful study.

Response: We thank the reviewer for their excellent suggestions which helped us improve our study dramatically. Especially their idea to support our suggestions through the inclusion of metaproteomic data helped us better understand the future utility of our study.

Reviewer #2 (Remarks to the Author):

The submitted manuscript has substantially improved through the first round of revisions and should be accepted for publication with a few additions. Below are previously identified concerns, how they were addressed by the authors, and additional comments.

(1) The original paper did not provide sufficient evidence for the translational or functional insights that tRNA-seq analyses may provide over existing 16S rRNA seq methods. The updated manuscript includes new analysis on (a) tRNA anticodon-based taxonomy and (b) the relationship of a tRNA modification with microbiome protein expression that better highlight the utility of the new tool.

a. tRNA anticodon-based taxonomy analysis: Analysis of tRNA^{Glu} (CUC) and tRNA^{Glu} (UUC) revealed that the occurrence of TTC genes varied among bacterial classes and, in particular, can be used to distinguish *Lactobacillaceae* in mice fed HF and LF diets, and even allowed for species level analysis of these bacteria.

b. Proteomics and tRNA modification analysis: The authors integrated analysis from a previously published proteomics study looking at mice fed HF and LF diets, where they performed mass spectrometry and identified proteins with significantly different expression between the two diet groups. It is impressive that the authors were able to validate some of their findings from the tRNA-seq experiment with metabolomics data from an independent study. I think there needs to be more discussion on the comparisons being made (see below), but otherwise think that it is a convincing example of how tRNA-seq and metaproteomics could be utilized in future studies.

Response: We thank the reviewer for their summary of the improvements in our revised manuscript.

(2) The authors clarified appropriate details regarding sample sizes and collection, sequencing

depth, reference database composition, and metric calculations. While the sample sizes for used (3 HF, 3 LF) were small, they are reconfirming microbiome differences from previous work and using it as a pilot to show that their method is feasible and has important implications. Figures were also sufficiently updated for clarity. Some additional points that remain to be clarified include:

a. Page 21- “The day 43 experimental condition most closely resembles the experimental condition of our tRNA-seq experiment”- Why? What were the major differences in experimental conditions (if any) that could impact the findings? In the tRNA-seq experiment, mice were fed diets for four weeks (28 days) before sample collection—in the MetaPro-IQ approach, mice were fed a HF or LF diet for 43 days, and sampled on days 0, 14, 29, and 43—please clarify why didn’t you use the day 29 data.

Response: Among the day 0, 14, 29, 43 meta-proteomics samples, only day 29 and 43 showed differences in body weight for HF and LF fed mice. We analyzed the meta-proteomics data for both day 29 and 43 and found very similar results. The day 29 data are now presented in Supplementary Fig. 10, and a brief description is added on page 16.

b. Page 33- You mention Trimmomatic v0.32 was used to “remove primers, adapters, or any other low-quality sequences.” Please clarify the cutoff for a low-quality sequence (did you use default parameters or specify something different?)

Response: We used the default parameters, and now added this in the Methods section.

c. Page 37- “We obtained the taxonomic assignment for each of the 849 proteins using BLASTp”- What parameters (percent identity, evalue, etc) were used?

Response: This information is now provided in the Method section (under “Analysis of the microbiome metaproteomics data”). For example, among the BLASTp hits, 98.9% have e-value <e-14, 96.8% have >50% identity.

* Although minor additions, these details are especially important for clarity and reproducibility.

(3) Sufficient information was added to the discussion discussing the limitations and best-use cases of the methods described. I also think that the name change of the article better reflects the contents of the paper.

Response: We agree with the reviewer, and thank them for their careful evaluation of our study for clarity and reproducibility.

Reviewer #3 (Remarks to the Author):

In this revised manuscript, the authors have responded to reviewer comments with clarifications and new data from a gut microbiome proteomics publication. While the manuscript has been

significantly improved in terms of clarity, accuracy, and scholarship, the central argument that tRNA sequencing offers advantages for microbiome analysis than the traditional 16S rRNA gene sequencing is still not proven. The authors still have not demonstrated that tRNA-seq analysis offers advantages over 16S rRNA gene sequencing for quantitative taxonomic analysis of the microbiome or for identifying bacteria above the level of genus.

The authors have now expanded the manuscript to include a published gut microbiome metaproteomics analysis of the changes in the microbiome caused by high- and low-fat diets. The proteomics data was analysed in terms of the frequencies of amino acid usage in proteins differentially regulated by the diet. They coordinated this amino acid data with codon frequencies arising from the tRNA-seq analysis. While this is a very interesting analysis of microbiome proteomics and genomics, the proteomics data from another mouse study performed at a different institution two years ago cannot be compared to tRNA-seq data from the present study. Even though the same mouse background was used in both studies (C57BL/6), it is now well established in the literature that gut microbiomes differ dramatically from institution to institution, and within an institution at different times. Even with mice purchased from the same vendor at the same time but housed in different cages will have different microbiomes. Further, a different mouse chow was used for the high- and low-fat diet in the two studies. It is simply not credible to compare the published proteomics data to the present tRNA-seq data. If the proteomics data had been obtained from the animals used to perform tRNA-seq analysis, then these would be extremely exciting studies.

The results showing how tRNA sequences and tRNA modifications distinguish high- and low-fat diets are clear and novel. However, given that other sequencing-based approaches, including 16S analysis and metagenomics, could distinguish the dietary states, it is not clear that the manuscript rises to the level of novelty and impact expected for Nature Communications.

Response: We are very thankful for the reviewer for their time, careful reading of our analyses, and for their insights. We respectfully disagree with some of their comments, and would like to respond for collegiality and closure.

The reviewer statement “it is simply not credible to compare the published proteomics data to the present tRNA-seq data” is inconsistent with the ideal of generality that research findings should not be confined within a single experiment, method, and/or group of investigators. While this ideal is not always achievable, our analysis of amino acid and codon pair effects with tRNA-seq and meta-proteomics derived from two different microbiome studies precisely argues for the generality of our observations. In fact, our ability to confirm our observations using data from another independent study gives more credibility to tRNA-seq approaches.

The reviewer also raises concerns regarding 16S rRNA gene and tRNA-seq comparisons. For instance, they suggest “the central argument that tRNA sequencing offers advantages for microbiome analysis than the traditional 16S rRNA gene sequencing is still not proven”. We find this suggestion rather confusing since in fact we did *prove* that tRNA sequencing can offer advantages for microbiome analyses. This is evidenced by our functional insights into a

microbial system uniquely through tRNA-seq, and by the fact that 16S rRNA gene amplicon surveys carry no information regarding such aspects of naturally occurring microbial populations. That said, our manuscript neither attempts to *invalidate* appropriate uses of 16S rRNA gene amplicon surveys, nor presents tRNA-seq as a *competing* method for those tasks. In fact, our manuscript ends with this sentence: “[tRNA-seq] offers a **complementary** approach to the sequencing of 16S rRNA gene amplicons (...) for broader characterization of environmental microbiomes”. Every new method comes with advantages and disadvantages. tRNA-seq offers high-throughput characterizations of modification fractions of individual tRNA species which provides a glimpse into the cell state as these fractions can be associated with protein expression dynamics as suggested by our new analyses. This aspect of tRNA-seq is indeed advantageous to 16S rRNA gene amplicons.

The reviewer also suggests that “[it is] not demonstrated that tRNA-seq analysis offers advantages over 16S rRNA gene sequencing for quantitative taxonomic analysis of the microbiome”. This suggestion incorrectly implies that 16S rRNA gene sequencing represents the ‘quantitative truth’. The reliance on primers with limited breadth, occurrence of variable number of rRNA operons in a given genome across taxa, the number of chimeric sequences as a function of the number of PCR cycles, the significant impact of target amplicon length on the assessment of microbial membership and abundance, and the habitat-dependent efficacy of primers are some of major limitations of 16S rRNA gene amplicon studies that are broadly documented in the literature, and should serve as reminders to refrain from taking quantitative estimates from this strategy as the ground truth. One of the most recent examples for the failure of primer-based strategies to quantify environmental microbes was demonstrated in Apprill et al. (doi:10.3354/ame01753), as the initial Earth Microbiome Project primers did not amplify SAR11, the most abundant microbial member of most surface ocean samples. Also, recently our group demonstrated the broader shortcomings of primer-based strategies to quantify environmental microbes (doi:10.1038/s41564-018-0176-9). In contrast, tRNA-seq does not rely on any primers of specific sequences, and does offer insights into the translational machinery of naturally occurring microbial communities. The quantitative power of tRNA-seq is indeed an open question, and we hope that future studies will address this aspect of this new strategy critically and systematically. However, with its well-documented limitations, 16S rRNA gene amplicons do not necessarily represent the high bar tRNA-seq must be benchmarked against.

The reviewer also suggests that “other sequencing-based approaches, including 16S analysis and metagenomics, could distinguish the dietary states”. This is correct. Yet, the purpose of our study was not to distinguish dietary states, but to validate that tRNA-seq as a method *could* distinguish between states that are known to be distinct, while offering additional insights that other approaches could not offer.

Once again we thank all reviewers for their invaluable suggestions and critiques throughout the review process.